



# The ocean's biological and preformed carbon pumps in future steady-state climate scenarios

Benoît Pasquier[1], Mark Holzer[1], and Matthew A. Chamberlain[2]

[1]School of Mathematics and Statistics, University of New South Wales, Sydney, NSW, Australia
[2]Commonwealth Scientific and Industrial Research Organisation, Hobart, TAS, Australia

**Correspondence:** Benoît Pasquier (b.pasquier@unsw.edu.au) and Mark Holzer (mholzer@unsw.edu.au)

**Abstract.** The future of the marine carbon cycle is vitally important for climate and the fertility of the oceans. However, predictions of future biogeochemistry are challenging because a myriad of processes needs parameterization and the future evolution of the physical ocean state is uncertain. Here, we embed a data-constrained model of the carbon cycle in steady circulations that correspond to perpetual 2090s conditions as simulated for the RCP4.5 and RCP8.5 scenarios. Focusing on
steady-state changes from preindustrial conditions allows us to capture the response of the system on all timescales, not just on the sub-centennial timescales of typical transient simulations. We find that biological production experiences only modest declines because the reduced nutrient supply by a more sluggish future circulation is counteracted by warming-stimulated growth. Organic-matter export declines by 15–25 % due to reductions in both biological production and export ratios, the latter driven by warming-accelerated shallow respiration and reduced subduction of dissolved organic matter. The future biological
pump cycles a 30–70 % larger regenerated inventory accumulated over longer sequestration times, while preformed DIC is shunted away from biological utilization to outgassing. We develop a conceptually new partitioning of preformed DIC to quantify the ocean's preformed carbon pump and its future changes. Near-surface paths of preformed DIC become more important in the future as weakened ventilation isolates the deep ocean. Thus, while regenerated DIC cycling becomes slower in the future, preformed DIC cycling speeds up for inventory changes of similar magnitude.





## 1 Introduction

The ocean carbon cycle is a key control on future climate (e.g., Revelle and Suess, 1957; Broecker, 1982; Arias et al., 2021) and the sustained biological productivity of the ocean is essential for future food security (e.g., Golden, 2016; FAO, 2018). Anthropogenic forcing of the climate system will have profound effects on the future carbon cycle and has already begun to impact ocean biogeochemistry (e.g., Riebesell et al., 2009). In recent decades dissolved oxygen has declined (e.g., Helm et al.,

2011; Whitney et al., 2007), the ocean is acidifying (e.g., Doney et al., 2009), and the upper ocean has warmed and stratified (e.g., Li et al., 2020). The question of how the ocean carbon cycle will evolve in the future is thus of intense interest and has stimulated numerous studies (for a small selection see, e.g., Sarmiento and Le Quéré, 1996; Sarmiento et al., 1998; Matear and Hirst, 1999; Plattner et al., 2001; Riebesell et al., 2009; Bopp et al., 2013; Bernardello et al., 2014; Ito et al., 2015; Hauck et al., 2015; Moore et al., 2018; Henson et al., 2022; Wilson et al., 2022; Liu et al., 2023).

Carbon sequestration in the ocean occurs through two mechanisms (Volk and Hoffert, 1985): (i) organic-matter production in the surface ocean is followed by the export of primarily biogenic particles to depth (the biological pump) and (ii) $CO_2$ is highly soluble in seawater (Weiss, 1974) and newly dissolved inorganic carbon (DIC) is physically transported into the ocean interior with the circulation (the solubility pump). In response to climate change, both the biological and solubility pumps are expected to change. This will have major consequences for the radiative forcing of the atmosphere (without the biological

pump preindustrial atmospheric $CO_2$ concentrations would have been about 200 ppm higher; see, e.g., Volk and Hoffert, 1985; Holzer et al., 2021b), and for future food security as biological production is expected to be diminished (e.g., Kwiatkowski et al., 2020).

For the 21st century and beyond there is broad agreement among models of the fifth and sixth phases of the Coupled Model Intercomparison Project (CMIP5 and CMIP6) that the ocean will sequester more carbon (Bopp et al., 2013; Ito et al., 2015;

Hauck et al., 2015; Henson et al., 2022; Wilson et al., 2022; Liu et al., 2023). There is also broad agreement that despite future reductions in productivity and export (e.g., Bopp et al., 2013), the biological pump will cycle a larger pool of regenerated carbon in the future as a slowing overturning circulation increases the residence time of carbon in the deep ocean (e.g., Ito et al., 2015; Wilson et al., 2022; Liu et al., 2023).

However, the future evolution of the ocean carbon cycle remains uncertain (Riebesell et al., 2009; Henson et al., 2022)

for many reasons: Future radiative forcing depends on uncertain socioeconomic factors (Meinshausen et al., 2020), numerous physical and complex biogeochemical processes need to be parameterized and calibrated (e.g., Gent et al., 1995; Kriest, 2017; Kriest et al., 2020), and ocean models must be properly spun up so that future changes are not biased by initial drift (e.g., Irving et al., 2021). Moreover, typical centennial simulations are unable to probe the response of the carbon cycle to future conditions on all its natural timescales, which exceed a millennium (Primeau, 2005; Holzer and Primeau, 2010; Holzer et al.,

2021a). Recognizing the importance of the slow deep circulation for carbon cycling, some studies have explored the response of the ocean's carbon pumps to the year 2300 (e.g., Moore et al., 2018; Liu et al., 2023), but even that only probes a fraction of the full spectrum of timescales that will shape the long-term evolution of ocean biogeochemistry.





The central question of our work here is: What would the equilibrium steady-state ocean carbon cycle be if the ocean circulation and thermodynamic state at the end of the 21st century were frozen in time? Steady states are advantageous because they probe all the timescales of the system and avoid the complications of transience. To reduce uncertainty due to poorly constrained biogeochemical parameters, we employ a data-constrained model of the carbon cycle with optimized parameters (Pasquier et al., 2023). To avoid the computational costs of model spin-up, we directly solve for equilibrium using steady-state ocean circulation transport matrix models built from the ocean circulation as simulated for the 2090s under the RCP4.5 and RCP8.5 future scenarios by the Australian Community Climate and Earth System Simulator (ACCESS1.3; Bi et al., 2013). (Representative Concentration Pathways RCP4.5 and RCP8.5 represent intermediate and worst-case scenarios for future global warming (Meinshausen et al., 2011).)

To comprehensively track all carbon through the ocean, we quantify future change in terms of the biological carbon pump and in terms of what we call the preformed carbon pump. A quantification of the preformed pump is made possible by a novel partition of preformed DIC according to its sources and sinks. The preformed pump can be regarded as the abiotic solubility pump, although the latter term is often used specifically for the subduction of DIC driven by solubility gradients (e.g., Volk and Hoffert, 1985). The preformed pump defined here allows us to track the transformation of regenerated into preformed DIC as old regenerated carbon resurfaces, and the transformation of preformed into regenerated DIC by tracking the surface pathways to biological utilization. In this way we quantify the timescales and flow rates (the "plumbing") of not just the biological pump but also of the preformed pump, as well as the interaction between these pumps and the atmosphere and how these change in the future under idealized steady-state scenarios.

We find that biological production is resilient with warming-stimulated plankton growth counteracting a diminished nutrient supply. The biological pump becomes more sluggish in the future due to an overall slowdown in circulation, which leads to a larger regenerated DIC pool being pumped at reduced rate. Concurrently, the slower circulation and reduced ventilation of the future tend to isolate the deep ocean. This dramatically *decreases* the bulk residence times of preformed DIC as its transport becomes surface-intensified. Increased preformed flow rates, driven by increased atmospheric $p\mathrm{CO}_2$, reroute DIC away from biological utilization underscoring the subtle interplay between the biotic and abiotic parts of the carbon cycle.

## 2 Methods

### 2.1 Ocean Circulation Models

To build our circulation models, we use physical ocean states from ACCESS1.3 climate-model simulations (Bi et al., 2013) for the preindustrial ocean and for two future climate scenarios. For the preindustrial ocean, we use the 1990s average of the circulation, thermodynamic, and forcing fields from the "historical" ACCESS1.3 runs submitted to CMIP5 (Taylor et al., 2012). (By the 1990s there have been relatively minor physical changes from preindustrial conditions compared to the large future changes analyzed here.) For the future ocean states, we similarly use the 2090s average for the ACCESS1.3 CMIP5 runs for the RCP4.5 and RCP8.5 scenarios. These correspond to intermediate (RCP4.5) and worst-case (RCP8.5) greenhouse-gas levels





(Meinshausen et al., 2011). For our biogeochemical model, we prescribe atmospheric $CO_2$ mixing ratios according to these
RCP scenarios at 278, 536, and 886 ppm for the preindustrial, RCP4.5, and RCP8.5 biogeochemical states analyzed below.

The ocean's advective–diffusive flux-divergence operator is discretized on the numerical grid and organized into "transport
matrices" for each state following the approach of Chamberlain et al. (2019). The horizontal advective fluxes across grid-
cell faces are taken from the averaged ACCESS1.3 fields and the vertical fluxes are calculated from mass conservation by

integrating up from the seafloor. To speed up numerical solutions, we coarse-grain the grid by lumping 2×2 horizontally neigh-
boring grid cells as done by Pasquier et al. (2023). Using the original resolution without re-optimizing parameters degrades
the biogeochemical model's match with observations with little improvement in finer-scale features. The coarse-grained grid
has a nominal horizontal resolution of 2°×2° (finer in latitude near the equator) and 50 depth levels with layer thicknesses
that increase from 10 m for the surface layer to 335 m for the deepest layer. We prescribe background horizontal and vertical

diffusivities of 500 $m^2 s^{-1}$ and $10^{-5} m^2 s^{-1}$, respectively, and a mixed-layer vertical diffusivity of 0.1 $m^2 s^{-1}$. (The transport
matrix for the preindustrial state is identical to the ACCESS-M matrix used by Pasquier et al. (2023).)

We use the transport matrices to embed our biogeochemical model (PCO2, Pasquier et al., 2023) in the corresponding
circulations. The transport matrices and atmospheric $CO_2$ concentrations are held fixed in time, and we solve for steady state
using an efficient iterative Newton solver. We thus calculate the equilibrium biogeochemical state under perpetual preindustrial

and 2090s RCP4.5 and RCP8.5 conditions. While this avoids spin-up issues, we emphasize that our solutions for the future
cannot be interpreted as predictions for the 2090s. Instead, our steady-state solutions allow us to probe all the system's natural
timescales by determining what the asymptotically long-term adjustment of the carbon cycle would look like if the circulations
were frozen in time.

## 2.2 Biogeochemistry Model

We use PCO2, a simple model of the ocean's carbon, nutrient, and oxygen cycles (Pasquier et al., 2023). Here, we only detail
the features and parameterizations directly relevant to the present work. Crucially, PCO2 mechanistically couples the carbon,
phosphorus, and oxygen cycles capturing important nonlinear interactions and feedbacks in the response of the biogeochemistry
to climate change. These include the effect of nutrient supply and temperature on biological production and the effect of
temperature and oxygen on bacterial respiration.

The parameters of the PCO2 model were objectively optimized for our preindustrial state against the observed concentra-
tions of phosphate ($PO_4$), DIC, oxygen, and total alkalinity, with DIC observations corrected for anthropogenic carbon. The
optimization tends to correct biases from the embedding circulation as well as biases from the biogeochemistry model (see
Section 4 for model caveats). In a nutshell, PCO2 makes the following simplifying approximations: Phosphate is the only
limiting nutrient, which avoids the complexities of the nitrogen cycle and micronutrients such as iron, and the marine ecosys-

tem is approximated by a single phytoplankton species with an implicitly parameterized mortality to avoid the considerable
complexities of higher trophic levels. These simplifications are justified a fortiori by the good fit to the observations.





We focus on DIC, which in steady state obeys

$$\mathcal{T}\,[\mathrm{DIC}] = -U + R + J_{\mathrm{atm}}, \tag{1}$$

where $\mathcal{T}\,[\mathrm{DIC}] = \nabla \cdot (\boldsymbol{u} - \mathbf{K}\,\nabla)\,[\mathrm{DIC}]$ is the flux divergence of DIC due to advection (with velocity $\boldsymbol{u}$) and eddy diffusion
(diffusivity tensor $\mathbf{K}$) and the local sources and sinks are on the right. The biological utilization rate per unit volume $U$ (further
detailed below) converts DIC to dissolved organic carbon (DOC), fast- and slow-sinking particulate organic carbon ($\mathrm{POC_f}$ and
$\mathrm{POC_s}$), or particulate inorganic carbon (PIC). Organic matter is then remineralized back to DIC through respiration at rates
$R_{\mathrm{DOC}}$, $R_{\mathrm{POC_f}}$, $R_{\mathrm{POC_s}}$, while PIC is redissolved to DIC at rate $D_{\mathrm{PIC}}$. In Eq. (1), $R$ denotes the total DIC regeneration rate,
i.e., $R = R_{\mathrm{DOC}} + R_{\mathrm{POC_f}} + R_{\mathrm{POC_s}} + D_{\mathrm{PIC}}$. In addition, DIC also has local sources or sinks $J_{\mathrm{atm}}$ through $CO_2$ exchange with
the atmosphere (described further below).

The biological DIC uptake rate $U$ is nonlinearly colimited by temperature, light, and nutrient availability, the latter being
additionally modulated by variable C:P uptake stoichiometry parameterized here in terms of phosphate concentration. Specifi-
cally we parameterize $U$ as

$$U = \underbrace{\frac{p_{\max}}{\tau}\,e^{\kappa T}}_{\beta}\,\underbrace{\left(\frac{\mathrm{PAR}}{\mathrm{PAR} + k_I}\right)^2}_{\lambda}\,\underbrace{\left(\frac{[\mathrm{PO_4}]}{[\mathrm{PO_4}] + k_{\mathrm{P}}}\right)^2\,\frac{1}{m\,[\mathrm{PO_4}] + b}}_{\alpha}, \tag{2}$$

where $\tau = 30\,\mathrm{d}$ is a growth timescale, $p_{\max} = 23.4\,\mu\mathrm{M}$ is a scale for phytoplankton concentration, $\kappa = 0.063\,\mathrm{K}^{-1}$ sets the
$e$-folding temperature for growth and mortality, and $k_I = 10\,\mathrm{W\,m}^{-2}$ and $k_{\mathrm{P}} = 3.14\,\mu\mathrm{M}$ are the half-saturation constraints for
photosynthetically active radiation (PAR) and $\mathrm{PO_4}$. The last fraction in Eq. (2) is the C:P uptake ratio parameterized in terms
of $[\mathrm{PO_4}]$ with slope $m = 6.9\,\mathrm{mmolP\,molC}^{-1}\,\mu\mathrm{M}^{-1}$ and intercept $b = 6.0\,\mathrm{mmolP\,molC}^{-1}$ (Galbraith and Martiny, 2015). (See
Table 1 of Pasquier et al. (2023) for the values of all model parameters.) In Eq. (2) we grouped terms into factors $\alpha$ (temperature-
related), $\lambda$ (light-related), and $\alpha$ (nutrient-related). With these definitions, the future-minus-preindustrial change $\Delta U$ can be
decomposed into contributions from the changes in each of these factors by writing $U + \Delta U = (\beta + \Delta\beta)\,(\lambda + \Delta\lambda)\,(\alpha + \Delta\alpha)$.
(Throughout, $\Delta X$ denotes the future-minus-preindustrial change in $X$ with $X$ being the preindustrial value.)

At the surface, carbon enters and exits the ocean through $CO_2$ air–sea exchange following the parameterization of Wan-
ninkhof (2014). Specifically, in Eq. (1), $J_{\mathrm{atm}} = w_0\,K_0\,(p\mathrm{CO}_2^{\mathrm{atm}} - p\mathrm{CO}_2^{\mathrm{ocn}})/z_0$ is the source/sink of $[\mathrm{DIC}]$ due to air–sea
exchange in the surface layer, where $w_0$ is the gas-transfer velocity, $K_0$ is the $CO_2$ solubility, $p\mathrm{CO}_2^{\mathrm{atm}}$ is the atmospheric
partial pressure of $CO_2$ at the sea surface, $p\mathrm{CO}_2^{\mathrm{ocn}}$ is the seawater equivalent partial pressure, and $z_0$ is the thickness of the
top model layer ($K_0$ and $p\mathrm{CO}_2^{\mathrm{ocn}}$ are computed using the MATLAB CO2SYS package; Lewis and Wallace, 1998; van Heuven
et al., 2011). Note that in our model carbon can only enter or exit the ocean through air–sea exchange so that in steady state
$\int J_{\mathrm{atm}}(\boldsymbol{r})\,\mathrm{d}^3\boldsymbol{r} = 0$.





### 2.3 Tracking preformed DIC from its sources to its sinks

We partition the DIC concentration into its usual preformed and regenerated components, but explicitly identify the sources and sinks of preformed DIC. Because the re-emergence of regenerated DIC into the euphotic zone is one of the sources of preformed DIC, we first consider regenerated DIC.

To track regenerated DIC, we label DIC during regeneration in the aphotic zone (mask $\Omega_{\mathrm{aph}} = 1$ in the aphotic zone and $0$ otherwise) and immediately unlabel it on entry into the euphotic zone (mask $\Omega_{\mathrm{eup}} = 1 - \Omega_{\mathrm{aph}}$). This unlabelling is conveniently accomplished by fast relaxation to zero with timescale $\tau_0 = 1\,\mathrm{s}$. The regenerated DIC concentration $C_{\mathrm{reg}}$ thus obeys

$$\mathcal{T}\,C_{\mathrm{reg}} = \Omega_{\mathrm{aph}}\,R - \Omega_{\mathrm{eup}}\,C_{\mathrm{reg}}/\tau_0. \tag{3}$$

$C_{\mathrm{reg}}$ is then straightforwardly partitioned according to regeneration mechanism (DOC, $POC_{\mathrm{f}}$, or $POC_{\mathrm{s}}$ respiration, or PIC dissolution) by replacing $R$ in Eq. (3) with the corresponding respiration or dissolution rate. The flow rates, residence times, and pathways of the mechanism-partitioned $C_{\mathrm{reg}}$ are then used to quantify the "plumbing" of the biological pump as in the work of Pasquier et al. (2023).

To track preformed DIC, we define a new labelling tracer that is allowed to roam over the entire ocean, including the euphotic zone, and for which we diagnose explicit euphotic sources and sinks. This contrasts sharply with the traditional approach where concentrations in the surface ocean (taken as the euphotic zone in many models or above the maximum mixed-layer depth in data-based analyses) are defined as preformed and then propagated into the ocean interior (e.g., Ito and Follows, 2005). Our preformed labelling tracer has the exact same concentrations as traditionally defined preformed DIC, but identifying its sources and sinks makes it possible to quantify the transport of preformed DIC within the surface ocean, which is not possible with the traditional approach.

Preformed DIC has three sources and two sinks. The sources are gross $CO_2$ ingassing at rate $J_{\mathrm{atm}}^{\downarrow} = w_0\,K_0\,p\mathrm{CO}_2^{\mathrm{atm}}/z_0$, regeneration within the euphotic zone at rate $\Omega_{\mathrm{eup}}\,R$, and emergence of aphotically regenerated DIC into the euphotic zone. When aphotically regenerated DIC enters the euphotic zone, its regenerated label is replaced by the preformed label so that the corresponding rate of labelling preformed DIC ("newly" preformed) is equal to the rate of unlabelling regenerated DIC given by $\Omega_{\mathrm{eup}}\,C_{\mathrm{reg}}/\tau_0$ in Eq. (3). The sinks are biological utilization, removing the preformed label at rate $U$, and gross outgassing, removing the preformed label at rate $J_{\mathrm{atm}}^{\uparrow} = w_0\,K_0\,p\mathrm{CO}_2^{\mathrm{ocn}}/z_0$. The preformed DIC concentration $C_{\mathrm{pre}}$ thus obeys

$$\mathcal{T}\,C_{\mathrm{pre}} = J_{\mathrm{atm}}^{\downarrow} + \Omega_{\mathrm{eup}}\,R + \Omega_{\mathrm{eup}}\,C_{\mathrm{reg}}/\tau_0 - J_{\mathrm{atm}}^{\uparrow} - U. \tag{4}$$

Casting the equation for preformed DIC in this way has major advantages over the traditional boundary-value approach. For the first time, we will be able (i) to partition preformed DIC according to its source and sink mechanisms, (ii) to track the transport of DIC through the surface ocean, and (iii) to quantify bulk flow rates of preformed DIC without the complications of diffusive one-way fluxes being singular (further discussion in Section 4).

To partition $C_{\mathrm{pre}}$ according to specified source origin ($s$) and sink destination (loss, $l$), we employ linear labelling tracers (e.g., Holzer et al., 2014; Pasquier and Holzer, 2018; Holzer and DeVries, 2022, details in Appendix A). This provides the





preformed concentration $C_{\mathrm{pre}}^{s \to l}$, which we use to quantify the "plumbing" of the ocean's preformed carbon pump and how it changes in the future.

## 3  Results

### 3.1  The future biological pump

The key components of the biological carbon pump are organic-matter production, export of organic matter, and sequestration of regenerated DIC in the aphotic interior. We now examine how each of these components change from their preindustrial values and what the main mechanisms are that drive these changes.

#### 3.1.1  Changes in organic-matter production

Organic-matter production in the euphotic zone, modelled here by Eq. (2) as DIC uptake $U$, is a key metric of ocean health and its future is of great importance for food security (e.g., Costello et al., 2020). The future circulation of our states is known to be more sluggish (Holzer et al., 2020) and the question is how this will affect future production.

Production appears to be remarkably resilient: The globally integrated production merely decreases by 8 % and 12 % for the RCP4.5 and RCP8.5 scenarios, respectively. Given the large physical and thermodynamic changes in our future scenarios, this resilience points to strong compensations between competing mechanisms. The nutrient supply does decline as expected from the more sluggish circulation: The global euphotic phosphate inventories decline by 12 and 19 % for the two scenarios (Fig. B1a–e). These declines are enhanced by increased Southern Ocean trapping (e.g., Primeau et al., 2013) due to the slower future circulation and decreased ventilation. The decreased nutrient supply reduces biological carbon uptake despite increases in C:P uptake ratios, as parameterized following Galbraith and Martiny (2015). This is counteracted by warming, which exponentially enhances nutrient and carbon uptake rates. The euphotic zone warms globally by 1.5 °C and 2.7 °C (Fig. C1) for RCP4.5 and RCP8.5 although the North Atlantic contains a patch of prominent cooling (e.g., Caesar et al., 2018).

Production declines because of lower nutrient supply despite warming-enhanced growth. To quantify the drivers of production change $\Delta U$, we decompose it algebraically into contributions from changes in nutrient limitation ($\Delta\alpha$), temperature growth factor ($\Delta\beta$), and light limitation ($\Delta\lambda$) (see Eq. (2); Fig. 1). The contributions from nutrient limitation only are $\beta\lambda\Delta\alpha = -18\,\mathrm{PgC\,yr^{-1}}$ and $-28\,\mathrm{PgC\,yr^{-1}}$ for RCP4.5 and RCP8.5, while the corresponding contributions from the temperature factor only are $\Delta\beta\lambda\alpha = +11\,\mathrm{PgC\,yr^{-1}}$ and $+22\,\mathrm{PgC\,yr^{-1}}$. The spatial correlations between $\Delta\alpha$ and $\Delta\beta$ (Fig. 1 purple bars) are negative as $\Delta\alpha$ and $\Delta\beta$ are of opposite sign and thus reinforce the production decrease. The other terms in the decomposition are less than about $1\,\mathrm{PgC\,yr^{-1}}$ in magnitude. Our finding that Southern Ocean nutrient trapping and reduced nutrient supply to the euphotic zone are the primary drivers of production decline is consistent with projections for the next few centuries (e.g., Moore et al., 2018).




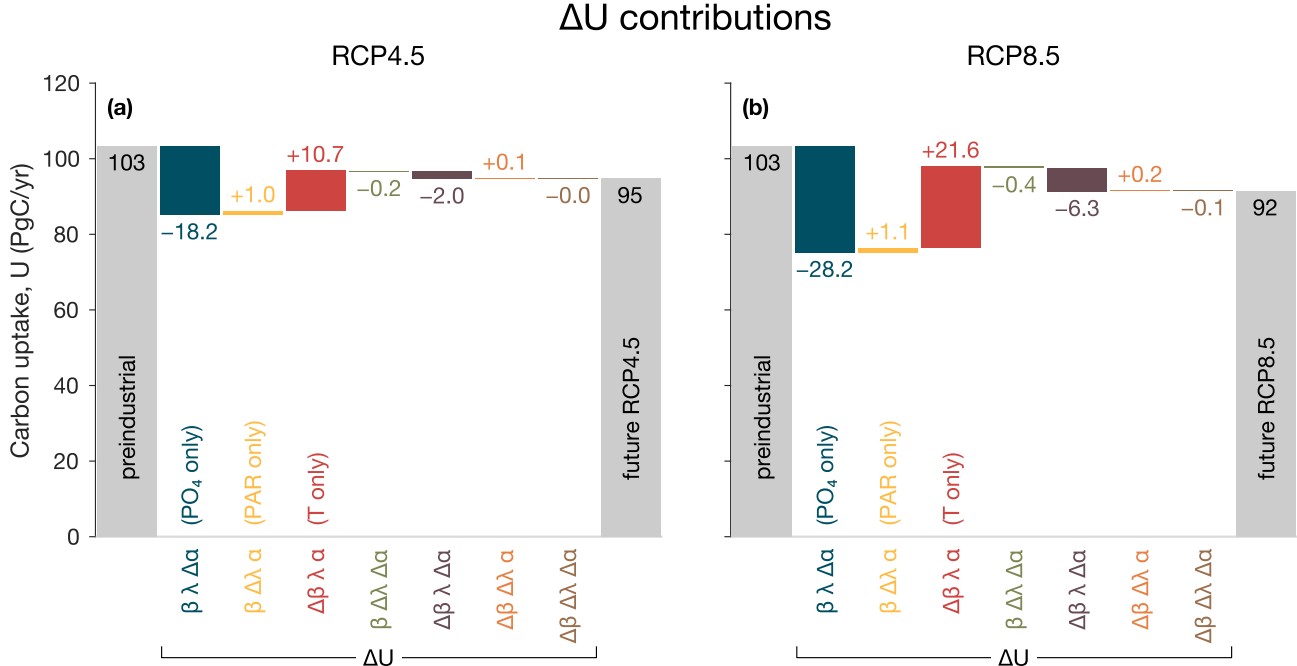

**Figure 1.** (a) Carbon uptake $U$ for preindustrial conditions (leftmost grey bar) and the RCP4.5-based future state (rightmost grey bar). The colored bars show the future-minus-preindustrial changes $\Delta U$ due to changes in the factors $\alpha$ (nutrients), $\lambda$ (light), and $\beta$ (temperature) (see Eq. (2)). These changes are plotted as a waterfall chart with each contribution (each bar) starting where the previous one ends. The values of the contributions are indicated in units of $PgC\,yr^{-1}$. (b) As (a) but for RCP8.5.

### 3.1.2 Changes in export ratios

The export ratio $f = J_{ex}/U$, where $J_{ex}$ is export production, is a measure of the efficiency of export and is useful for under-standing future changes in organic-matter export. $f$ generally decreases in the future (Fig. D2) with production-weighted global means reduced from a preindustrial value of 0.32 to 0.30 and 0.28 for RCP4.5 and RCP8.5. While export ratios are expected to decline overall with warming, it is understood that locally many factors other than temperature can play an important role (e.g., Laws et al., 2000, 2011; Cael and Follows, 2016). Because of the nonlocal and nonlinear coupling of the mechanisms that set export ratios in our model, a formal decomposition into factors of changes $\Delta$ is not useful.

Warming stimulates euphotic POC respiration (decreasing export ratios) but causes particles to sink faster through less viscous water (increasing export ratios). For POC, the more rapid sinking does not fully compensate the increased respiration rates (at least for our optimized parameters), so that the net effect is an overall reduction in POC export ratio (consistent with, e.g., Cael and Follows, 2016). For PIC (only dissolution, no respiration), the effect is faster sinking increasing the PIC export ratio, but this has little impact because the preindustrial PIC export ratio is already close to 1. Global euphotic deoxygenation (Fig. B1f–j) is driven by reduced oxygen solubility and reduced photosynthesis (except in the Weddell and Ross Seas due





to future sea-ice loss), which slows euphotic POC respiration thereby increasing POC export ratios. DOC export ratios are

expected to decrease with the circulation slowdown because DOC is exported by water transport.

### 3.1.3 Changes in export production

We now consider changes in carbon export $J_{ex}$ itself. $J_{ex}$ is the rate with which organic carbon exported at a given location is respired in the aphotic interior (e.g., Primeau et al., 2013; Kwon et al., 2022), which is a robust metric of the biological pump. Figure 2 shows $J_{ex}$ for the preindustrial state and for the two future scenarios along with the corresponding future-

minus-preindustrial change, $\Delta J_{ex}$. Globally integrated, $J_{ex}$ decreases by 14 % and 24 % in the RCP4.5 and RCP8.5 scenarios, respectively. This overall decrease in export production is driven by the combined action of changes in temperature, PAR, and euphotic nutrient and oxygen concentrations, themselves driven by changes in ocean circulation and air–sea exchange.

Large reductions in $J_{ex}$ (Fig. 2g–j) occur approximately in regions of large preindustrial production with the Southern Ocean being most affected. In the Ross Sea, a large decrease in $[PO_4]$ results in a significant decrease in uptake and export. This is

because the unrealistic deep mixed layer of the preindustrial ACCESS circulation (Bi et al., 2013), which brings nutrient-rich deep waters to the surface, disappears in the future states (Fig. C2). By contrast, in the Weddell Sea, export production increases despite the disappearance of the unrealistically deep mixed layer because retreating sea ice allows for additional photosynthesis. There are also patches of export increases of similar magnitude in the Sea of Okhotsk, in the mid-latitude southern Atlantic and Indian Ocean, and in the North Atlantic. The patterns of $\Delta J_{ex}$ broadly correlate with those of $\Delta U$ but

some notable differences (Figs. 2 and D1) suggests that changes in organic-matter production alone do not suffice to explain changes in export production.

To quantify the role of changes in organic-matter production ($\Delta U$) and changes in export ratio ($\Delta f$) in shaping future export production, we decompose $\Delta J_{ex}$ into contributions from $\Delta U$ and $\Delta f$ and their spatial correlation. We find that in the global mean changes in export ratio alone $U \Delta f$ contribute about 50 % more to $\Delta J_{ex}$ than changes in production alone $f \Delta U$ (Fig 3).

The cross-term contribution $\Delta f \Delta U$ captures the spatial correlation between $\Delta U$ and $\Delta f$ and is positive but about an order of magnitude smaller (Fig 3c,f). The declines in future export production are thus driven by both changes in production and export ratios, but the latter make larger contributions globally, particularly for the RCP8.5 scenario.

The spatial patterns of the contributions $U \Delta f$ and $f \Delta U$ to the export-production changes are very different in character, with $U \Delta f$ being negative almost everywhere (Fig. 3a,d) and $f \Delta U$ having both signs (Fig. 3b,e). In the Northern Hemisphere,

many features of $U \Delta f$ mirror (with opposite sign) the patterns of the surface temperature and oxygen changes (Figs. C1 and B1i,j) suggesting important contributions from both. This is not the case in the Southern Ocean, where changes in export ratios are likely driven by changes in DOC export driven in turn by the large circulation changes. The pattern of $f \Delta U$ shows that $\Delta U$ drives both decreases (negative) and increases (positive over about 40 % of the ocean) in export production. The changes in organic-matter production itself $\Delta U$ are likely due to the competing effects of changes in nutrient concentrations, temperature,

and light as documented for the global integral above (section 3.1.1). $f \Delta U$ is largest in magnitude at high latitudes where it is dominated by the pattern of $\Delta U$ with zonal bands of alternating sign indicating meridional shifts in organic-matter production.







**Figure 2.** (a) Map of export production, $J_{ex}(r)$, vertically integrated, in the preindustrial state. (b) Zonal integral of (a). (c)–(d) As (a)–(b) but for the future RCP4.5-based state. (e)–(f) As (c)–(d) but for RCP8.5. (g)–(j) As (c)–(f) but for $\Delta J_{ex}$.





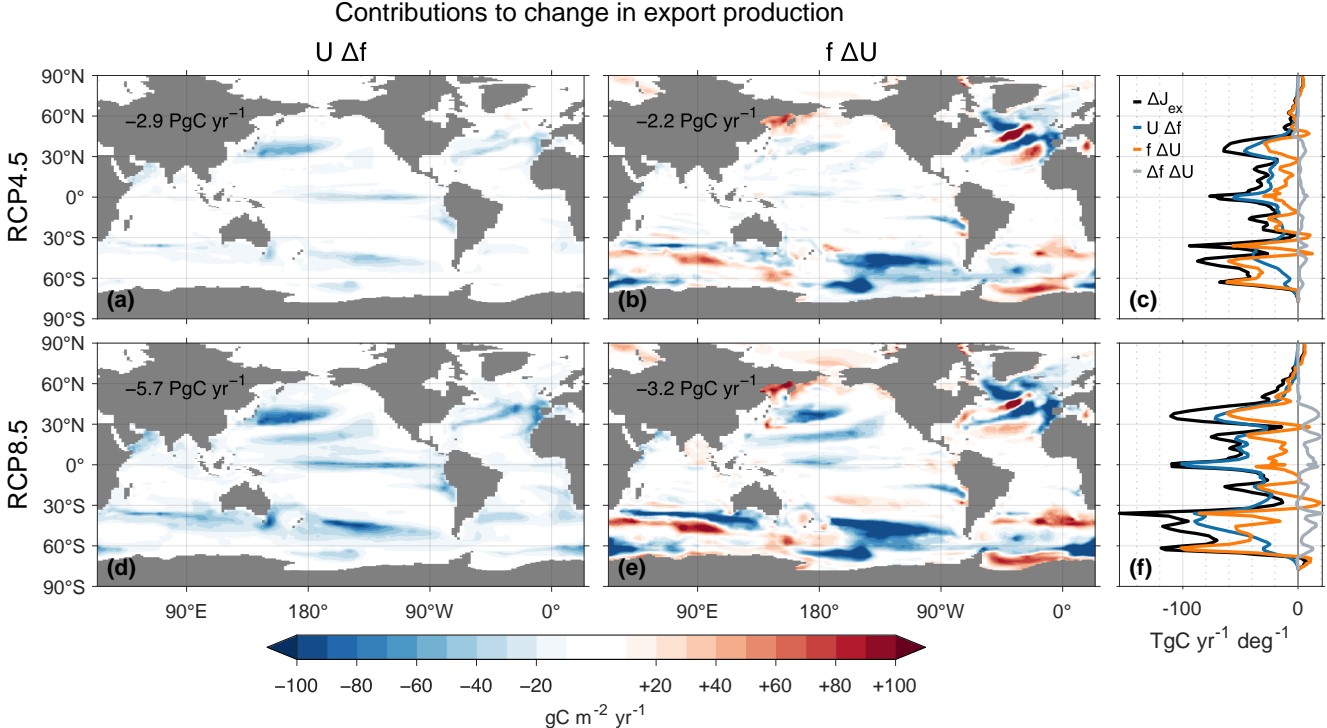

**Figure 3.** (a,b,d,e) Maps of vertically integrated contributions from changes in export ratio $U \Delta f$ (a,d) and carbon uptake $f \Delta U$ (b,e) to the changes in export production $\Delta J_{\mathrm{ex}}$, in the RCP4.5 (a,b) and RCP8.5 (d,e) scenarios. (c,f) Corresponding zonal integrals of $U \Delta f$ (blue), $f \Delta U$ (orange), and $\Delta J_{\mathrm{ex}}$ (black). Also shown is the zonal integral of $\Delta f \Delta U$ (gray).

Overall, the contributions $U \Delta f$ tend to dominate near the equator while the contributions $f \Delta U$ from changes in organic-matter production are more important at high latitudes, where $f$ is larger (Fig. D2).

## 3.2 Global changes in carbon sequestration

The sequestration strengths of the biological and preformed carbon pumps are conveniently quantified by the sizes of the regenerated and preformed inventories. Table 1 lists the total, regenerated, and preformed DIC inventories ($\mu_{\mathrm{tot}}$, $\mu_{\mathrm{reg}}$, and $\mu_{\mathrm{pre}}$) for the preindustrial and future states. The ocean's total carbon inventory increases by 5 % and 8 % for the RCP4.5 and RCP8.5 scenarios, respectively. Contributions to the changes from regenerated and preformed DIC are on the same order despite being driven by distinct mechanisms, which we will now explore in detail.

To trace the pathways of regenerated DIC, we plot in Fig. 4 the zonal mean $C_{\mathrm{reg}}$ and its future-minus-preindustrial change, $\Delta C_{\mathrm{reg}}$. The zonal-mean $\Delta C_{\mathrm{reg}}$ (Fig. 4j–o) shows large increases throughout the entire deep ocean of up to about 100 µM and 200 µM for the RCP4.5 and RCP8.5 scenarios, respectively. The increases are most prominent in the deep Southern Ocean, where a slower circulation increasingly traps regenerated DIC. (For the RCP4.5-based scenario, the Southern Ocean increase



**Table 1.** Total, regenerated, and preformed carbon inventories, in PgC (rounded to hundreds of PgC).

| State | $\mu_{\text{tot}}$ | $\Delta\mu_{\text{tot}}$ | $\mu_{\text{reg}}$ | $\Delta\mu_{\text{reg}}$ | $\mu_{\text{pre}}$ | $\Delta\mu_{\text{pre}}$ |
|---|---|---|---|---|---|---|
| Preindustrial | 35500 | | 1900 | | 33600 | |
| Future RCP4.5 | 37200 | +1700 | 2500 | +600 | 34800 | +1200 |
| Future RCP8.5 | 38400 | +2900 | 3300 | +1400 | 35100 | +1500 |

is limited in the Atlantic sector because the unrealistically deep MLD, which short-circuits Southern Ocean nutrient trapping
(Pasquier et al., 2023), has not entirely subsided.) The slight decreases at low latitudes and thermocline depths are likely due
to decreased tropical organic-matter production.

The regenerated DIC inventory $\mu_{\text{reg}}$ is governed by the product of globally integrated export production $\Phi_{\text{ex}} = \int J_{\text{ex}}(\boldsymbol{r})\,\mathrm{d}^3\boldsymbol{r}$
and bulk sequestration time $\Gamma_{\text{reg}} = \mu_{\text{reg}}/\Phi_{\text{ex}}$ for which the respired exported organic matter is allowed to accumulate in the
aphotic ocean (e.g., DeVries et al., 2012; Holzer et al., 2021b). We find that the fraction of DIC that is regenerated increases
from its preindustrial values of 5.4 % to 6.7 % (a 25 % increase) for RCP4.5 and to 8.5 % (a 60 % increase) for RCP8.5 (Fig. 5).
We will now show that these increases are driven primarily by changes in the circulation pathways that return regenerated DIC
to the euphotic zone.

The pie charts of Fig. 5 show the fractional contributions to the global regenerated DIC inventory from the export of DOC,
$\text{POC}_{\text{s}}$, $\text{POC}_{\text{f}}$, and PIC, for the preindustrial state and each future scenario. The contributions from each of these export mech-
anisms are remarkably impervious to change with future-minus-preindustrial differences within 1 %–3 % for both RCP4.5
and RCP8.5. This is surprising *a priori* because DOC, $\text{POC}_{\text{s}}$, $\text{POC}_{\text{f}}$, and PIC do not remineralize at the same location and
particularly not at the same depth.

To demonstrate that bulk sequestration time is the key control on the biological pump we partitioned the regenerated DIC
inventory into separate pools according to regeneration mechanism and calculated the flow-rate through each pool (equal to
the volume-integrated regeneration rate). Each such pool may be considered to be a "pipe" in the "plumbing" of the biological
pump, with the bulk sequestration time for each pipe simply being the ratio of inventory in the pipe to the corresponding flow
rate. These pipes are depicted as horizontal bars in Figure 5 whose length, width, and area quantify the bulk sequestration
time (yr), flow rate ($\text{PgC yr}^{-1}$), and inventory (PgC), respectively (as done previously for the optimized preindustrial state by
Pasquier et al., 2023).

Figure 5 shows that for each regeneration mechanism, export-production rates decline and sequestration times increase
consistent with an overall slow down of the circulation and longer mean water re-exposure time (the time for water at a given
interior location to return to the surface ocean; Fig. C3). Across all mechanisms, export production rates decline by 8–16 % and
11–27 % for RCP4.5 and RCP8.5. These export changes affect $\text{POC}_{\text{s}}$ the most because the pattern of its export-ratio changes
correlate well with preindustrial production (see Figs. D3c,g and D1) and PIC the least because PIC dissolution in our model is
independent of temperature, oxygen, or circulation. Across all mechanisms, re-exposure times increase by 40–50 % and 100–
160 % for RCP4.5 and RCP8.5. Reexposure times of DIC regenerated from slow-sinking $\text{POC}_{\text{s}}$ are the most affected, likely



**Figure 4.** (a)–(c) Zonal mean $C_{\mathrm{reg}}$ for the preindustrial state in the Atlantic (a), Pacific (b), and Indian Ocean (c). (For all zonal means shown in this work, the Atlantic basin excludes the Gulf of Mexico and the Caribbean, and the Pacific basin excludes the Sea of Japan so that the averages are more cleanly interpretable.) (d)–(f) As (a)–(c) but for the future RCP4.5-based state. (g)–(i) As (d)–(f) but for RCP8.5. (j)–(o) As (d)–(i) but for $\Delta C_{\mathrm{reg}}$.



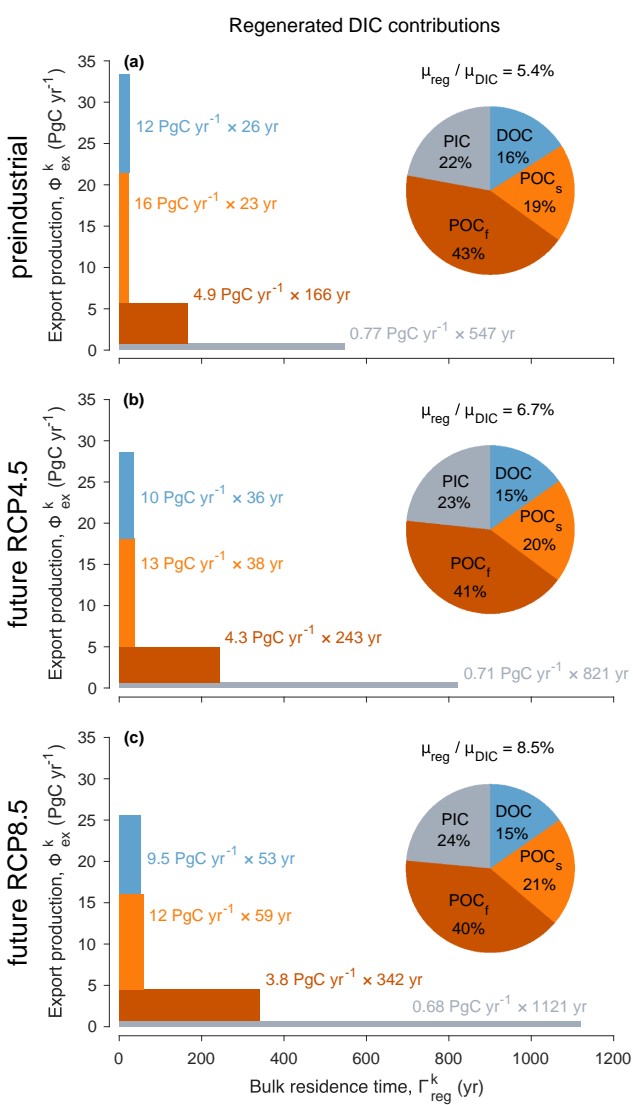

**Figure 5.** (a) Regenerated DIC contributions from each mechanism $k =$ DOC, POC$_s$, POC$_f$, and PIC (blue, orange, red, and grey) for the preindustrial state represented as both a pie chart and a bar chart. The pie chart quantifies the fractional contribution $\mu_{reg}^k / \mu_{reg}$. Pump sequestration strength, as quantified by $\mu_{reg} / \mu_{DIC}$, is indicated above the pie chart. Each bar can be interpreted as a pipe flowing from left to right and whose area represents the regenerated DIC inventory, $\mu_{reg}^k$, which equals globally integrated export production, $\Phi_{ex}^k$ (pipe width, $y$-axis) multiplied by bulk residence time, $\Gamma_{reg}^k$ (pipe length, $x$-axis). (b) As (a) but for the future RCP4.5 state. (c) As (b) but for RCP8.5.

because POC$_s$ is respired at an average depth of about $200\,\mathrm{m}$, where the relative changes in re-exposure times are the largest (Fig. C3).

The effect of increased bulk sequestration times wins over the effect of decreased export production rates. This is shown in

Table 2, where the change in each pipe's regenerated DIC inventory $\Delta\mu_{reg}$ is formally decomposed into contributions from





**Table 2.** Contributions to the change in the regenerated carbon inventory $\Delta\mu_{\mathrm{reg}}^k$ from changes in bulk residence time $\Gamma_{\mathrm{reg}}^k$ and export production $\Phi_{\mathrm{ex}}^k$ for each pump mechanism $k$ (values in PgC rounded to two significant digits).

| Scenario | Mechanism $k$ | $\Delta\mu_{\mathrm{reg}}^k$ | $=$ | $\Phi_{\mathrm{ex}}^k \Delta\Gamma_{\mathrm{reg}}^k$ | $+$ | $\Gamma_{\mathrm{reg}}^k \Delta\Phi_{\mathrm{ex}}^k$ | $+$ | $\Delta\Gamma_{\mathrm{reg}}^k \Delta\Phi_{\mathrm{ex}}^k$ |
|---|---|---|---|---|---|---|---|---|
| RCP4.5 | total | +580 | | +950 | | −240 | | −120 |
| | DOC | +73 | | +130 | | −38 | | −16 |
| | POC$_s$ | +140 | | +230 | | −59 | | −38 |
| | POC$_f$ | +210 | | +380 | | −110 | | −52 |
| | PIC | +160 | | +210 | | −35 | | −18 |
| RCP8.5 | total | +1400 | | +2200 | | −390 | | −460 |
| | DOC | +190 | | +320 | | −63 | | −66 |
| | POC$_s$ | +320 | | +570 | | −98 | | −160 |
| | POC$_f$ | +490 | | +860 | | −180 | | −190 |
| | PIC | +340 | | +440 | | −48 | | −51 |

$\Delta\Gamma_{\mathrm{reg}}$ and $\Delta\Phi_{\mathrm{ex}}$, including cross terms. Overall, regenerated inventories increase by 25–40 % and 60–90 % across all regeneration mechanisms for the RCP4.5 and RCP8.5 scenarios. In both scenarios, the circulation changes quantified here by $\Delta\Gamma_{\mathrm{reg}}$ are the dominant control: The increase in regenerated inventory due to longer reexposure times alone ($\Phi_{\mathrm{ex}}\Delta\Gamma_{\mathrm{reg}}$) is ∼4–6 times larger than the decrease due to smaller export production alone ($\Gamma_{\mathrm{reg}}\Delta\Phi_{\mathrm{ex}}$). In addition, the cross term $\Delta\Gamma_{\mathrm{reg}}\Delta\Phi_{\mathrm{ex}}$, which captures the spatial correlations between $\Delta\Phi_{\mathrm{ex}}$ and $\Delta\Gamma_{\mathrm{reg}}$, makes contributions on the same order as $\Gamma_{\mathrm{reg}}\Delta\Phi_{\mathrm{ex}}$, underlining the highly non-linear response of the biological pump in both scenarios.

### 3.3 The preformed carbon pump

To assess the future state of the biological pump, it is useful to place it in the context of overall carbon sequestration by also considering the preformed DIC pool (e.g., Ito et al., 2015). Our approach differs conceptually from previous efforts to quantify the abiotic ocean carbon cycle, which have typically focused on the solubility pump relative to saturated surface conditions (e.g., Volk and Hoffert, 1985). Here, we simply consider the preformed counterpart of the biological pump in terms of our new preformed DIC tracer. This allows us to quantify both interior and euphotic DIC pathways and how these change in the future.

[t]

We partition the ocean's preformed DIC into pools that connect specified sources of preformed DIC (ingassing, euphotic regeneration, or emergence of aphotically regenerated DIC) to specified sinks (outgassing or biological uptake). Each pool may again be considered to be a "pipe" of the ocean's preformed DIC pump with the flow rate through each pool determining the corresponding bulk residence time. To the best of our knowledge, this is the first time that preformed DIC has been partitioned in this way (see Methods and Appendix A for mathematical details).







**Figure 6.** Preformed DIC pipes for (a) the preindustrial state, (b) the future RCP4.5-based state, and (c) the future RCP8.5-based state, partitioned according to source (left) and sink (right). The area, width ($y$-axis), and length ($x$-axis) of each $s{\rightarrow}l$ pipe represent the inventory $\mu_{\mathrm{pre}}^{s{\rightarrow}l}$ (white text), flowrate $\Phi_{\mathrm{pre}}^{s{\rightarrow}l}$ (black text at the end of each pipe), and bulk residence time $\Gamma_{\mathrm{pre}}^{s{\rightarrow}l}$, respectively. For comparison, the dashed rectangle in the bottom left corner of each panel indicates the total regenerated DIC pipe (all mechanisms summed/averaged from Fig. 5), also represented as export $\Phi_{\mathrm{ex}}$ (width) × bulk residence time $\Gamma_{\mathrm{reg}}$ (length).

 

Figure 6 shows the preformed DIC inventory $\mu_{\mathrm{pre}}$, partitioned into "pipes" that connect each source $s$ to each loss $l$, for the
preindustrial and future states. As in Fig. 5, the inventory $\mu_{\mathrm{pre}}^{s \to l}$ (area) of each pipe is the product of globally integrated flow
rate $\Phi_{\mathrm{pre}}^{s \to l}$ ($y$-axis width) and bulk residence time $\Gamma_{\mathrm{pre}}^{s \to l}$ ($x$-axis length). When interpreting Figure 6 it is useful to keep in mind
the following constraints on the flow rates: (i) The emergence rate of aphotically regenerated DIC into the preformed pool is
equal to the export production $J_{\mathrm{ex}}$ as source and sink balance in steady state. (ii) The biological utilization rate of preformed
DIC is equal to the total nutrient uptake $U$ as euphotic DIC is preformed by definition. (iii) The euphotic regeneration rate is
equal to the rate of un-exported production $U - J_{\mathrm{ex}} = (1 - f)U$ in steady state. Given the novelty of these diagnostics, we first
take a look at the preindustrial state before examining future changes.

### 3.3.1  The preindustrial preformed pump

The (vertical) widths of the preformed pipes (Fig. 6) that terminate in biological utilization represent the rate with which
preformed DIC fuels photosynthetic organic-matter production. In the preindustrial ocean, the fraction of newly preformed DIC
destined for biological uptake is remarkably large at 60–65 % across all source processes. ("newly" preformed DIC denotes
preformed DIC at the time of labelling it as "preformed", i.e., at its source.) For instance, of the $63\,\mathrm{PgC\,yr^{-1}}$ ingassed from the
atmosphere, $38\,\mathrm{PgC\,yr^{-1}}$ support biological production, while the remainder outgasses without ever interacting with biology.
In terms of inventories, 60 % of all preformed DIC is in transit to biological uptake while the rest is destined for outgassing.
Conversely, of the total biological DIC uptake that occurs, 36 % is supplied by ingassing, 44 % by euphotic regeneration, and
20 % by upwelling of aphotically regenerated DIC.

To understand the transport pathways of preformed DIC better, we ask how much of it is able to enter the aphotic ocean.
To that end, we partitioned the DIC in each preformed pipe according to whether or not it enters the aphotic interior during
its source-to-sink transit (Eqs. (A9) and (A10)). This partition (not shown) revealed that regardless of source–sink pair more
than 90 % of preformed DIC that has been newly injected into the euphotic zone ("newly preformed") will eventually explore
the aphotic interior with a bulk first-contact time with the aphotic interior of less than half a year (roughly in agreement with
the findings of Bopp et al., 2015). The preformed DIC pipes thus consist almost entirely of pathways that probe the aphotic
interior.

Figure 6a also shows the bulk preindustrial residence times in each preformed DIC pipe, which range from 150 yr to 280 yr,
depending on pipe. In comparison, the bulk residence time for regenerated DIC is only about 60 yr, which might seem surprising
given that DIC regeneration occurs at depth. However, preformed DIC can roam over the euphotic zone for times on the order
of a century without being biologically utilized or outgassed, while regenerated DIC immediately loses its regenerated label
on contact with the euphotic zone. Furthermore, not all deep-to-surface paths are slow: Analysis of nutrient cycling in a data-
assimilated circulation (Pasquier and Holzer, 2016) showed that regenerated nutrients (and hence DIC) are likely to return to the
surface through short diffusive vertical pathways while deep regenerated DIC is predominantly transported into Circumpolar
Deep Water (CDW), which provides a relatively fast upwelling conduit back to the surface. For the ACCESS-M PCO2 model,
the CDW conduit is additionally short-circuited regionally in the Southern Ocean by the parent model's unrealistic deep mixing
(Fig. C2 and also Holzer et al., 2020).





**Table 3.** Contributions to the response of the preformed carbon inventory $\Delta\mu_{\text{pre}}^{s\rightarrow l}$ from changes in bulk residence time $\Delta\Gamma_{\text{pre}}^{s\rightarrow l}$ and changes in flow rate $\Delta\Phi_{\text{pre}}^{s\rightarrow l}$ (values in PgC and rounded to two significant digits).

| Scenario | Connection (source $s$ ⟶ loss $l$) | $\Delta\mu_{\text{pre}}^{s\rightarrow l}$ | $\Phi_{\text{pre}}^{s\rightarrow l}\Delta\Gamma_{\text{pre}}^{s\rightarrow l}$ | $\Gamma_{\text{pre}}^{s\rightarrow l}\Delta\Phi_{\text{pre}}^{s\rightarrow l}$ | $\Delta\Gamma_{\text{pre}}^{s\rightarrow l}\Delta\Phi_{\text{pre}}^{s\rightarrow l}$ |
|---|---|---|---|---|---|
| RCP4.5 | total (all ⟶ all) | +1200 | −7800 | +11000 | −2300 |
| | ingassing ⟶ outgassing | +6600 | −1200 | +9900 | −2100 |
| | ingassing ⟶ bio. uptake | +170 | −1900 | +2800 | −680 |
| | euphotic regen. ⟶ outgassing | +540 | −910 | +1800 | −370 |
| | euphotic regen. ⟶ bio. uptake | −3200 | −1700 | −2000 | +520 |
| | $C_{\text{reg}}$ emergence ⟶ outgassing | +2.4 | −740 | +940 | −190 |
| | $C_{\text{reg}}$ emergence ⟶ bio. uptake | −2900 | −1300 | −2100 | +510 |
| RCP8.5 | total (all ⟶ all) | +1500 | −15000 | +28000 | −12000 |
| | ingassing ⟶ outgassing | +12000 | −2300 | +26000 | −11000 |
| | ingassing ⟶ bio. uptake | −1000 | −3600 | +4700 | −2100 |
| | euphotic regen. ⟶ outgassing | +320 | −1700 | +3300 | −1200 |
| | euphotic regen. ⟶ bio. uptake | −4800 | −3000 | −3200 | +1400 |
| | $C_{\text{reg}}$ emergence ⟶ outgassing | −940 | −1600 | +1300 | −590 |
| | $C_{\text{reg}}$ emergence ⟶ bio. uptake | −4300 | −2600 | −3300 | +1600 |

### 3.3.2 The future preformed pump

In our perpetual future scenarios, the global preformed DIC inventory $\mu_{\text{pre}}$ increases from its preindustrial value by $\sim 1200\,\text{PgC}$
and $1500\,\text{PgC}$ for the RCP4.5 and RCP8.5 scenarios, respectively (Table 3). While this is only about $5\,\%$ of the total preformed inventory, the absolute changes are of the same order of magnitude as (and larger than) the changes in the regenerated DIC pool ($\sim 600\,\text{PgC}$ and $1400\,\text{PgC}$ for the two scenarios).

The plumbing of the preformed DIC pump is reshaped entirely in our future scenarios. Preformed DIC is rerouted from biological utilization to outgassing. The strong increase in gross outgassing rate (by roughly a factor of 2 and 3 for RCP4.5
and RCP8.5) reflects the corresponding factor of 2 and 3 increases in atmospheric $p\text{CO}_2$. The roles of reduced $\text{CO}_2$ solubility in warmer surface waters and changes in gas-exchange coefficients due to wind changes are dwarfed by the prescribed $p\text{CO}_2$ increases. The increase in outgassing rate reduces the fraction of newly preformed DIC destined to biological utilization from a preindustrial $60$–$65\,\%$ to $40$–$50\,\%$ and $30$–$35\,\%$ in the two future scenarios (ranges are given across source mechanisms). Similarly, the fraction of biological production supported by newly ingassed DIC increases from $35\,\%$ in the preindustrial era
to $50\,\%$ and $65\,\%$ in the future. Across all source–sink pairs, the ingassing-to-outgassing flow rates increase the most (roughly by factors 3 and 6 for RCP4.5 and RCP8.5) because of the dramatic change in atmospheric $p\text{CO}_2$.

The re-routing of preformed DIC points to the biological pump becoming *less* efficient in the future: Regenerated DIC is much less likely to pass another time through the biological pump before outgassing to the atmosphere. Specifically, only



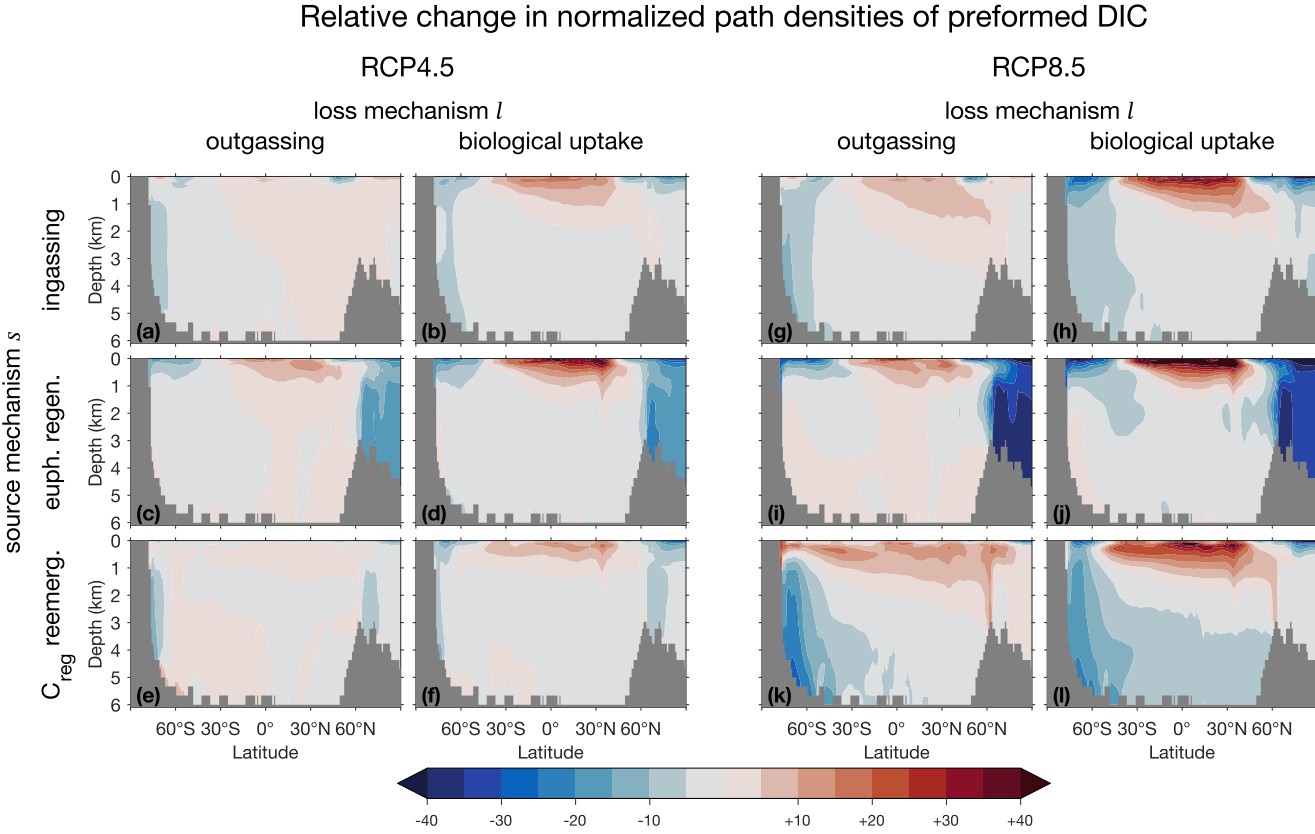

**Figure 7.** Zonal means of the change of the normalized path density of preformed DIC from each source mechanism $s$ (rows) to each loss mechanism $l$ (columns), $\Delta\left[C_{\mathrm{pre}}^{s\to l}/\bar{C}_{\mathrm{pre}}^{s\to l}\right]$, for the RCP4.5-based state (a)–(f) and the RCP8.5-based state (g)–(l). Note that the normalization by the global mean $\bar{C}_{\mathrm{pre}}^{s\to l}$ is necessary to uncover changes in pattern, given the large overall changes in the magnitude of $\bar{C}_{\mathrm{pre}}^{s\to l}$.

about 40 % (RCP4.5, 30 % for RCP8.5) of preformed DIC sourced from regenerated organic matter is destined to biological
utilization in the future compared to 60 % preindustrially. In the sense that the number of life-time passages through the
biological pump is a measure of carbon pump efficiency (Holzer et al., 2021b), the future biological pump becomes *less*
efficient. Using the vertical attenuation of the POC flux as a measure of pump efficiency (e.g., Buesseler et al., 2020; Volk and
Hoffert, 1985), we similarly find that the efficiency of POC transfer to 1000 m below the euphotic zone decreases by order 5 %
in the future. Hence, the pump becomes *stronger* in the sense that it sequesters a larger regenerated DIC pool in the future, but
it does so *less* efficiently in the sense of reduced export rate and lifetime biological pump passages.

A key feature of the future preformed pump is that the bulk residence times in its "pipes" are shorter than in the preindustrial
state. Preformed DIC pipes thus respond oppositely to regenerated DIC pipes: While the flow rate of regenerated DIC slows
down with the more sluggish future circulation, the flow rate of preformed DIC actually speeds up in the future: Bulk preformed



residence times across all the pipes decrease by 20–25 % and 40–50 % for the RCP4.5 and RCP8.5 scenarios. This at first
counterintuitive behavior can be understood in terms of the future deep ocean becoming more isolated due to weaker future
ventilation. Faster surface paths are thus expected to become more important contributors to the overall transport than the
slower deep paths. To confirm and quantify this, we calculated the path density (e.g., Holzer and Primeau, 2013; Pasquier and
Holzer, 2016) for source- and sink-partitioned preformed DIC. For a given source–sink pair this path density is simply the
concentration of DIC in transit from specified source to specified sink. Figure 7 shows the future-minus-preindustrial changes
in the global zonal-mean normalized preformed DIC path densities. (The path densities were normalized by the corresponding
global preformed DIC inventory to reveal changes in pattern.) For all source–sink pairs, the future path density becomes
surface-intensified with strongly increased upper-ocean transport away from high latitudes. This is consistent with changes
in the fraction of newly preformed DIC that will remain entirely within the euphotic zone before outgassing or biological
utilization (Eqs. (A9)–(A10); not shown), which increases from about 5 % in the preindustrial state to 8 % and 12 % in the
RCP4.5- and RCP8.5-based states.

## 4   Discussion

We have analyzed the ocean carbon cycle for idealized steady-state future conditions corresponding to a perpetual 2090s
ocean state as predicted for the RCP4.5 and RCP8.5 scenarios. Our approach tracks all carbon in the ocean by considering
the biologically pumped regenerated carbon pool, the preformed carbon pool, and the exchanges between these pools and the
atmosphere. By focusing on the steady-state equilibrium response of the carbon cycle, we gain insight into how the system
adjusts on all timescales, including the long millennial timescales of deep ventilation (e.g., Primeau and Holzer, 2006). This
is important to capture, e.g., strengthened Southern Ocean nutrient trapping, which is driven by circulation slowdown and
manifests in transient simulations only well after the 21st century (Moore et al., 2018). In contrast, 21st-century transient
simulations can only probe the response on centennial and sub-centennial timescales.

To track preformed DIC not only in the interior but also in the surface ocean, we introduced a conceptually new partition of
preformed DIC according to the sources that inject DIC into the euphotic zone and the sinks that remove it. This represents a
significant advance over the traditional view of preformed DIC as being merely the propagated surface DIC concentration (e.g.,
Ito and Follows, 2005), with which it is impossible to track preformed DIC transport in the surface ocean. Our new approach
allows us to paint a quantitative picture of the "plumbing" of the ocean's preformed carbon pump and how it connects the
atmosphere to the biological pump.

Our new partition of preformed tracers will be useful beyond the scope of the present work. For example, a key quantity
of interest in the ocean's nutrient cycles is the fraction of production in one region supported by the supply of nutrients that
are newly preformed in another region. This cannot be accurately quantified using the traditional concept of preformed tracers
because of the singular diffusive return flux of locally labelled preformed tracers (Hall and Holzer, 2003; Primeau and Holzer,
2006). Previous approaches have sidestepped this difficulty by either perturbing the system in the origin region and attributing
the nonlinear response elsewhere to contributions from the origin region (e.g., Sarmiento et al., 2004), using interior volumes to




label nutrient origin (e.g., Palter et al., 2010), or labelling nutrient origin according to last biological utilization or regeneration (e.g., Holzer and Primeau, 2013; Pasquier and Holzer, 2016). By contrast, our new partition of preformed tracers has non-diffusive sinks and is hence not subject to diffusive singularities making it the ideal tool for tracking preformed connectivity in the ocean.

A number of caveats must be kept in mind: (i) As previously emphasized, our future states are steady and thus not predictions of future transient evolution. However, our idealized steady states do capture the system's key responses to future change (discussed further below). (ii) Our circulation models have no seasonality and therefore cannot capture temporal covariances between physical, thermodynamic, and biological fluctuations (e.g., Riebesell et al., 2009). While this could be addressed with a cyclo-stationary model (e.g., Bardin et al., 2014), doing so would greatly increase complexity and computational cost. (iii) Most quantitative aspects of our results are likely model specific and imprinted to some extent by unrealistic circulation features like Antarctic bottom-water formation through deep mixing inherited from the parent climate model (ACCESS1.3; Bi et al., 2013; Pasquier et al., 2023). However, the qualitative link between changes and their driving mechanisms should be robust and model-independent. For example, mixed-layer shoaling and intensified nutrient trapping in more sluggish future circulations have been seen across a number of climate models (e.g., Liu et al., 2023). (iv) Our model may not capture potentially important effects from mechanisms that are not explicitly parameterized (e.g., Henson et al., 2022). For example, these include changes in community composition from adaptation and evolution (e.g., Boyd, 2015; Passow and Carlson, 2012; Doney et al., 2009; Lomas et al., 2022) or changes in nitrogen and/or iron limitation (e.g., Thornton et al., 2009; Jickells et al., 2005). However, at least some of our model's mechanistic shortcomings are partially compensated by having optimized biogeochemical parameters (Pasquier et al., 2023).

How do the steady-state responses analyzed here compare with the trends seen in transient simulations? On one hand, we would expect a closer correspondence with the long-term behavior of multi-century simulations, where the system's slow processes have had a chance to begin to assert themselves. On the other hand, even the long-term evolution of the ocean will likely be characterized by transience (Schmittner et al., 2008) and an overturning slowdown may eventually recover once dynamical equilibrium becomes re-established (e.g., Bi et al., 2001; Jansen and Nadeau, 2019). Simulation time is thus key when comparing to steady-state responses and one must be careful with the interpretation of transient simulations. For example, an analysis of a transient 21st-century circulation slowdown may interpret a decreased preformed DIC inventory as a weakening of the solubility pump (e.g., Raven and Falkowski, 1999; Liu et al., 2023). However, at steady-state, a more sluggish circulation is expected to enhance the solubility pump and increase the preformed DIC inventory by allowing larger spatial gradients to form (see, e.g., Murnane et al., 1999; Toggweiler et al., 2003; DeVries, 2022), which is an effect we observe here.

Our estimates of future DIC increases by 1700–2900 PgC roughly agree with estimates from multi-century simulations. For example, using 350-yr simulations with 2× and 4× preindustrial $p$CO$_2$, Sarmiento and Le Quéré (1996) found increased DIC inventories ranging 1000–2000 PgC. Similarly, in simulations with $p$CO$_2$ prescribed at 550 and 1000 µatm, Plattner et al. (2001) found increases ranging 1000–1500 PgC by the year 2500. More recently, Liu et al. (2023) found an increase of about 1000 PgC by 2300 for the RCP8.5 scenario. For CMIP6 models under the SSP2-4.5 and SSP5-8.5 scenarios, the increase in ocean carbon sequestration is about 400–500 PgC by the year 2100 (Liu et al., 2023), of which only about 100 PgC is regenerated DIC as



there has been insufficient time for it to accumulate at depth. (SSP$x$-$y$ refers to "Shared Socioeconomic Pathway" scenario $x$ (Riahi et al., 2017) where $y$ is the approximate radiative forcing for 2100 in $\mathrm{W\,m^{-2}}$, which nominally matches RCP$y$ (Arias et al., 2021).)

The qualitative agreement of our results with the long-term behavior of transient simulations underlines the common driving mechanisms that are at work. Across almost all CMIP5 and CMIP6 models (e.g., Bopp et al., 2013; Hauck et al., 2015; Kwiatkowski et al., 2020; Arora et al., 2020; Wilson et al., 2022; Liu et al., 2023), a slower future circulation reduces nutrient supply, which reduces production and export despite warming-accelerated growth, and increases the sequestration time of regenerated DIC. At the same time, higher future atmospheric $p$CO$_2$ drives more carbon into the ocean. While the PCO2 model

explicitly represents only three ($T$-dependence of respiration, $T$-dependence of viscosity, and O$_2$-dependence on respiration) of twelve export-controlling mechanisms identified by Henson et al. (2022), our results broadly agree with the corresponding expected future effects in terms of both magnitude and direction: The dominant effect is that warming reduces export through enhanced shallow respiration, followed by faster particle sinking in less viscous seawater and decreased deep respiration due to reduced oxygen. Our results also roughly agree with the changes in the controls on carbon export and biological utilization

identified by Boyd (2015). However, their one-dimensional water-column analysis cannot capture the effects of circulation changes, which are a key control on the future carbon cycle identified here, while our analysis cannot capture changes in plankton community composition identified as a key driver by Boyd. (PCO2 may implicitly capture the effects of such changes through its optimized nonlinear production parameterization.)

    Our findings challenge the hypothesis of Liu et al. (2023) that a future circulation slowdown will reduce the capacity of

the ocean to take up anthropogenic CO$_2$ on multi-century timescales. Our analysis shows that the circulation slowdown is the dominant driver of the large regenerated DIC increases in the long-term steady state through increased residence times in the deep ocean. Slower circulation also increases the preformed DIC inventory by increasing DIC gradients thereby increasing the efficiency of the solubility pump (e.g., Murnane et al., 1999; Toggweiler et al., 2003), but this effect is overwhelmed by the large future increases in atmospheric $p$CO$_2$. The bulk of the DIC inventory is located in the deep ocean and returns to the surface

with a distribution of transit times that has a mean exceeding a millennium close to the seafloor rather than a few centuries (e.g., Fig. C3 below or Primeau, 2005). The correlation for the year 2100 between carbon uptake and decreased overturning across CMIP6 models found by Liu et al. can therefore only capture the relatively short-timescale response of preformed DIC and the correlation may thus very well change sign by 2300 or a few centuries later as more time becomes available to fill abyssal waters with additional DIC, assuming that the slow circulation persists.

Our decomposition of the change in regenerated DIC inventory into contributions from changes in export and circulation is consistent with the results of Liu et al. (2023), but our analysis identifies important nonlinearities that appear to be absent in their simulation to the year 2300. Broadly consistent with our findings, they found that the circulation slowdown contributes an increase in $\mu_{\mathrm{reg}}$ below $2000\,\mathrm{m}$ that is roughly 5× larger than the reduction due to weakened export. However, in stark contrast with our finding of a large correlation between changes in regeneration and reexposure times (roughly 20–30 % of the

total change, see Table 1), Liu et al. argue that the interaction between export and circulation is negligible. That the nonlinear interaction terms would fortuitously collapse to zero because of model specifics or experimental design is unlikely. Based on





our analyses, it is more likely that this difference stems predominantly from the fact that our future steady states capture the response associated with much longer timescales, allowing for the system's nonlinear interactions to develop fully.

## 5   Conclusions

We investigated the steady-state response of the ocean's carbon pumps to idealized future scenarios. This was done by embedding a relatively simple biogeochemical model into the average ocean state as predicted by the ACCESS climate model for the 2090s under the RCP4.5 and RCP8.5 scenarios. We then solved for the steady state of the biogeochemistry keeping the ocean state and prescribed atmospheric $p$CO$_2$ fixed.

Most features of the carbon cycle's response are already manifest in the RCP4.5-based scenario. For many quantities, the
response for RCP8.5 is approximately twice that for RCP4.5 roughly tracking the change in atmospheric $p$CO$_2$. One way in which the system's nonlinearities were revealed is in the spatial correlations between key driving mechanisms, which are 3–8 times higher for RCP8.5 than for RCP4.5. These nonlinearities can either strengthen (e.g., biological uptake) or weaken (e.g., export production) the overall response.

Our analysis focused on the mechanisms that drive the response to climate change, brought to light here by applying powerful
novel diagnostics of carbon sequestration and transport made possible by the steady-state framework. We partitioned the ocean's DIC into regenerated and preformed pools, further partitioned according to sources and sinks. Each partitioned DIC pool, together with its source-to-sink flow rate defining the residence time in each pool, may be considered to be an advective–diffusive "pipe". These pipes may be thought of constituting the "plumbing" of the biological and preformed carbon pumps. Whereas preformed DIC is usually considered as determined by a concentration boundary condition, here we introduced a new
partitioning of preformed DIC that allows us to track not only its interior transport, but also its surface pathways. To the best of our knowledge, this made it possible for the first time to track DIC from its injection into the preformed pool (by ingassing, euphotic regeneration, or upwelling of regenerated DIC) to its exit from the preformed pool through outgassing or biological uptake.

Our main conclusions are:

1. Biological productivity is remarkably resilient in the face of large environmental change for our perpetual 2090s scenarios, with declines of merely ∼10 % even for RCP8.5. These declines are driven by reduced nutrient supply but partially compensated by surface warming, which exponentially enhances organic-matter production and the recycling of nutrients within the euphotic zone. The overall nutrient supply declines because of intensified Southern Ocean nutrient trapping and because decreased ventilation slows the resurfacing of nutrients from depth.

2. Export production declines by 14 % and 24 % in the RCP4.5- and RCP8.5-based steady-state scenarios. These declines are driven not only by reduced organic-matter production but also by reduced export ratios. The reduction in export ratios is driven primarily by enhanced shallow POC respiration due to surface warming and by decreased DOC export due to reduced ventilation. Decreased respiration from deoxygenation and faster POC sinking in less viscous warmer water tends to increase export ratios but is of secondary importance.





3. The future biological pump cycles a larger regenerated DIC inventory, but does so less efficiently. The regenerated DIC inventory increases by 30 % ($\sim$600 PgC) and 70 % ($\sim$1400 PgC) for the RCP4.5- and RCP8.5-based scenarios, respectively. These steady-state increases capture millennial response timescales and are driven primarily by reduced deep ventilation that increases the sequestration time in the aphotic ocean. This allows more regenerated DIC to accumulate at depth and to participate in Southern Ocean trapping. Reduced export production only compensates for about half of the increase in regenerated inventories, underlining the key importance of circulation changes. Biological pump efficiency is decreased in the sense that resurfacing regenerated DIC contributes less to organic-matter production.

4. The preformed carbon pump is completely replumbed in our future scenarios. Preformed DIC is largely rerouted from supporting biological production to outgassing primarily because dissolved $CO_2$ at the surface, and hence gross ingassing and outgassing, increase 2–3 times in approximate proportion to the prescribed atmospheric $p$CO$_2$ increases. Reduced biological production and decreased $CO_2$ solubility in warmer waters play only a secondary role. Because the inventory of preformed DIC in the euphotic zone increases only by a few percent due to carbonate buffering, increased ingassing and outgassing rates correspond to preformed DIC having shorter residence times in the euphotic zone. As a consequence, the fraction of newly preformed DIC that supports biological production shrinks from about 60 % in the preindustrial to roughly $\sim$40 % in the future. However, the fraction of biological production supported by newly ingassed DIC increases from a preindustrial 36 % to more than 50 % in the future, again because of the large increase in ingassing.

5. Preformed DIC is cycled more rapidly in our future scenarios despite the overall slower circulation. Across all source and sink mechanisms, the residence times of preformed DIC in the entire ocean decrease by 20–50 % in the future, while the global preformed inventory increases by only a few percent ($\sim$ 5 % or 1500 PgC for the RCP8.5 case), implying faster source-to-sink flow rates. This is driven by fast shallow preformed transport pathways becoming more important relative to slow deep pathways as reduced ventilation isolates the deep ocean. The fraction of newly preformed DIC that never enters the aphotic ocean interior roughly doubles from its preindustrial value of 5 % in both scenarios.

Our analysis reveals a complex multifaceted response of the carbon cycle to changes in ocean state, even for our relatively simple biogeochemistry model and idealized scenarios. On one hand, a more sluggish future circulation slows down biological cycling but the regenerated carbon inventory increases due to longer sequestration times. On the other hand, preformed DIC shoals with faster source-to-sink flow due to the increased isolation of the deep ocean. While the response of the biological and preformed pumps are driven by different mechanisms with widely different response timescales, the regenerated and preformed DIC inventories both increase by similar amounts when the full spectrum of response timescales is captured in steady state.

*Code and data availability.* The MATLAB code for this work will be made available upon acceptance of this article. The transport matrices were built from the historical, RCP4.5, and RCP8.5 ACCESS1.3 CMIP5 model runs available at https://esgf.nci.org.au/projects/esgf‐nci/. This output also includes temperature, salinity, photosynthetically available radiation (PAR), sea-ice, and wind fields.





## Appendix A: Preformed DIC source-to-loss partition

To partition preformed DIC according to source and loss mechanism, we consider the following linear labelling tracer equation whose solution is identical to that of Eq. (4):

$$(\mathcal{T} + l_{\mathrm{atm}} + l_{\mathrm{bio}})\, C_{\mathrm{pre}} = J^{\downarrow}_{\mathrm{atm}} + \Omega_{\mathrm{eup}}\, R + \Omega_{\mathrm{eup}}\, C_{\mathrm{reg}}/\tau_0, \tag{A1}$$

where $l_{\mathrm{atm}} = J^{\uparrow}_{\mathrm{atm}}/[\mathrm{DIC}]$ and $l_{\mathrm{bio}} = U/[\mathrm{DIC}]$ are equivalent linear loss rates for preformed DIC corresponding to the inverse timescales of local outgassing and biological uptake, respectively. ($\tau_0 = 1\,\mathrm{s}$ is a fast relaxation timescale.) Equation (A1) exploits the fact that labels are removed in proportion to their fractional abundance even when the removal process is nonlinear (e.g., Holzer and DeVries, 2022). The Green function for Eq. (A1) obeys

$$\big(\mathcal{T}_{\boldsymbol{r}} + l_{\mathrm{atm}}(\boldsymbol{r}) + l_{\mathrm{bio}}(\boldsymbol{r})\big)\, G_{\mathrm{pre}}(\boldsymbol{r}|\boldsymbol{r}') = \delta(\boldsymbol{r} - \boldsymbol{r}'), \tag{A2}$$

where $\delta(\boldsymbol{r} - \boldsymbol{r}')$ is the three-dimensional Dirac distribution and the subscript in $\mathcal{T}_{\boldsymbol{r}}$ indicates that the differential operator $\mathcal{T}$ is applied at point $\boldsymbol{r}$. The Green function $G_{\mathrm{pre}}(\boldsymbol{r}|\boldsymbol{r}')$ allows us to cleanly partition $C_{\mathrm{pre}}$ according to every source and sink. Specifically, the preformed DIC contributed by any source $s$ is given by

$$C^s_{\mathrm{pre}} = \int G_{\mathrm{pre}}(\boldsymbol{r}|\boldsymbol{r}')\, s(\boldsymbol{r}')\, \mathrm{d}^3\boldsymbol{r}', \tag{A3}$$

which in matrix form is equivalent to

$$\boldsymbol{C}^s_{\mathrm{pre}} = (\mathbf{T} + \mathbf{L}_{\mathrm{atm}} + \mathbf{L}_{\mathrm{bio}})^{-1}\, \boldsymbol{s} \tag{A4}$$

where $\mathbf{T}$ is the advective–diffusive transport matrix, $\mathbf{L}_{\mathrm{atm}}$ and $\mathbf{L}_{\mathrm{bio}}$ are diagonal matrices representing the local loss rates $l_{\mathrm{atm}}$ and $l_{\mathrm{bio}}$, and $\boldsymbol{s}$ is a column vector of the source $s$.

To quantify the fraction of $C_{\mathrm{pre}}$ destined for a given loss mechanism (outgassing or uptake), we take the adjoint Green function defined as the solution to

$$\big(\widetilde{\mathcal{T}}_{\boldsymbol{r}} + l_{\mathrm{atm}}(\boldsymbol{r}) + l_{\mathrm{bio}}(\boldsymbol{r})\big)\, \widetilde{G}_{\mathrm{pre}}(\boldsymbol{r}|\boldsymbol{r}') = \delta(\boldsymbol{r} - \boldsymbol{r}'), \tag{A5}$$

where $\widetilde{\mathcal{T}}$ is the adjoint of $\mathcal{T}$ with respect to the volume-weighted inner product. The fraction of preformed DIC destined for loss process $l$, where $l = l_{\mathrm{bio}}$ or $l_{\mathrm{atm}}$, is given by

$$f^l_{\mathrm{pre}} = \int \widetilde{G}_{\mathrm{pre}}(\boldsymbol{r}|\boldsymbol{r}')\, l(\boldsymbol{r}')\, \mathrm{d}^3\boldsymbol{r}'. \tag{A6}$$

In matrix form, Eq. (A8) becomes

$$\boldsymbol{f}^l_{\mathrm{pre}} = (\widetilde{\mathbf{T}} + \mathbf{L}_{\mathrm{atm}} + \mathbf{L}_{\mathrm{bio}})^{-1}\, \boldsymbol{l}, \tag{A7}$$

where $\widetilde{\mathbf{T}} = \mathbf{V}^{-1}\mathbf{T}^{\mathsf{T}}\mathbf{V}$ is the adjoint of $\mathbf{T}$ with respect to the volume-weighted inner product ($\mathbf{V}$ is a diagonal matrix of grid-box volumes) and $\boldsymbol{l}$ is the vector representing the grid-point values of $l$.





The preformed DIC concentration due to source mechanism $s$ and destined for loss mechanism $l$ is thus given by

$$C_{\text{pre}}^{s\to l} = f_{\text{pre}}^l \, C_{\text{pre}}^s. \tag{A8}$$

The inventory of the $s\to l$ preformed DIC pipe, i.e., the amount of preformed DIC in transit from source $s$ to loss $l$, is given by $\mu_{\text{pre}}^{s\to l} = \int C_{\text{pre}}^{s\to l}(\boldsymbol{r}) \, \mathrm{d}^3\boldsymbol{r}$. Similarly, the global flowrate of the $s\to l$ preformed DIC pipe is given by $\Phi_{\text{pre}}^{s\to l} = \int f_{\text{pre}}^l(\boldsymbol{r}) \, s(\boldsymbol{r}) \, \mathrm{d}^3\boldsymbol{r}$. The bulk transit time is then simply given by $\Gamma_{\text{pre}}^{s\to l} = \mu_{\text{pre}}^{s\to l} / \Phi_{\text{pre}}^{s\to l}$.

To better diagnose the physical pathways of preformed DIC, we further partition the DIC in the preformed pipes into DIC that remains in the euphotic zone throughout the preformed lifetime and DIC that explores the aphotic interior. To this end, we

introduce a labelling tracer for euphotic-only DIC, $C_{\text{pre}}^{\text{eup}}$, which is governed by

$$(\mathcal{T} + l_{\text{atm}} + l_{\text{bio}} + l_{\text{aph}}) \, C_{\text{pre}}^{\text{eup}} = J_{\text{atm}}^{\downarrow} + \Omega_{\text{eup}} \, R + \Omega_{\text{eup}} \, C_{\text{reg}} / \tau_0, \tag{A9}$$

which is similar to Eq. (A1) but where $l_{\text{aph}} = \Omega_{\text{aph}} / \tau_0$ additionally unlabels any preformed DIC that leaves the euphotic zone. Taking the difference with the full preformed DIC from Eq. (A1) gives the concentration $C_{\text{pre}}^{\text{aph}}$ of preformed DIC that will explore the aphotic interior and obeys

$$(\mathcal{T} + l_{\text{atm}} + l_{\text{bio}}) \, C_{\text{pre}}^{\text{aph}} = l_{\text{aph}} \, C_{\text{pre}}^{\text{eup}}. \tag{A10}$$

In practice, Eqs. (A9) and (A10) are again solved in matrix form.





## Appendix B: Changes in tracer fields

Nutrient supply (here $PO_4$) is the dominant driver of the response of production (Fig. 1), while oxygen concentrations control the respiration rates in the euphotic zone and thus export ratios. Figure B1 shows the euphotic-zone mean $[PO_4]$ and $[O_2]$ in the preindustrial and future states, along with the corresponding change. Phosphate concentrations decrease almost everywhere, with a particularly strong decrease near the Weddell Sea and the Ross Sea, where the mixed layer that was unrealistically deep preindustrially has shoaled considerably (Fig. C2). For euphotic-zone $[O_2]$, we also see a global decline except in the Weddell and Ross seas, where oxygen was strongly undersaturated preindustrially because of the unrealistic MLDs.





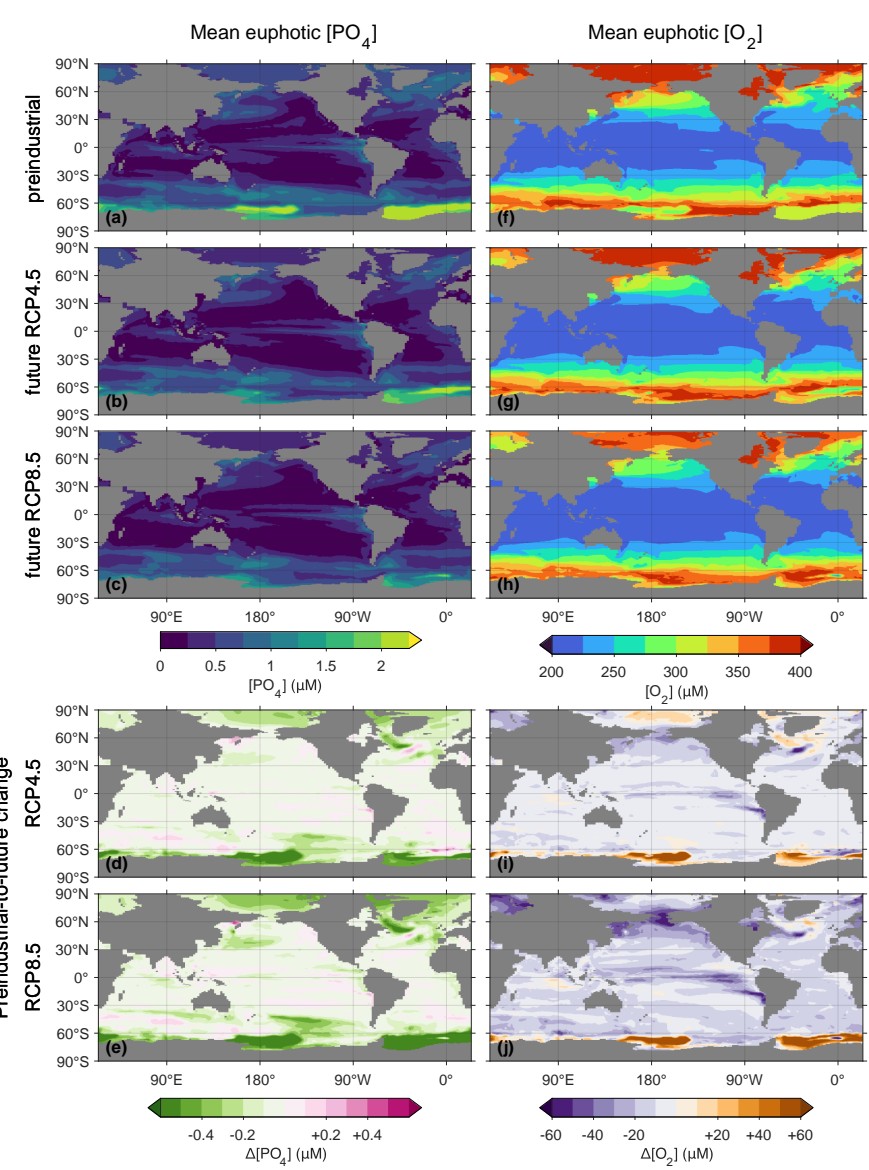

**Figure B1.** (a) Mean euphotic $[PO_4]$ for the preindustrial state. (b) As (a) for the RCP4.5-based state. (c) As (b) for RCP8.5. (d)–(e) As (b)–(c) but for $\Delta[PO_4]$. (f)–(j) As (a)–(e) but for $[O_2]$.



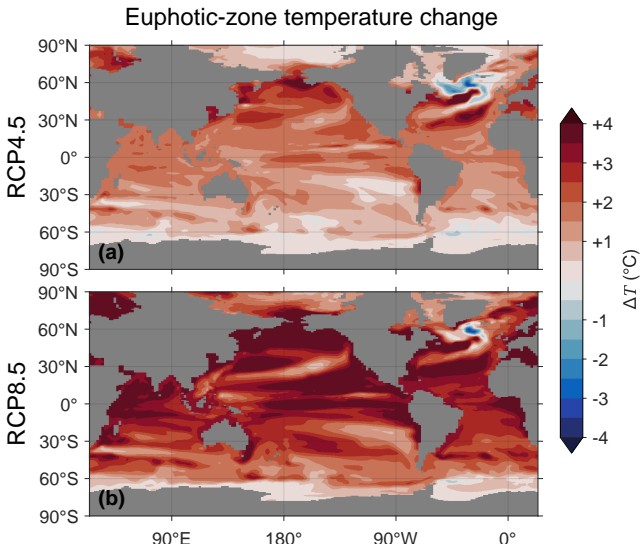

**Figure C1.** (a) Change in vertical mean euphotic temperature, $\Delta T$, for the RCP4.5 scenario. (b) As (a) but for RCP8.5.

### Appendix C:  Thermodynamic Fields and Circulation Features

Through its control on phytoplankton and bacterial growth rates, temperature change is a key driver of production and export changes. Figure C1 shows the changes in euphotic-zone temperature from the preindustrial values for the RCP4.5 and RCP8.5 scenarios. (The temperature fields used here are decadal means for the 1990s and 2090s of the parent ACCESS1.3 model, consistent with our methodology for extracting the circulations.) Temperature strongly increases almost everywhere except in the highest latitudes of the Southern Ocean and in the subpolar North Atlantic "cold blob" where temperatures decrease

(thought to be driven partly by the Atlantic overturning circulation slowdown; see, e.g., Cheng et al., 2022).

The mixed layer plays a key role in supplying nutrients to the euphotic zone. Figure C2 shows the mixed layer depths (MLDs) in the preindustrial and future states. (As for temperature, the MLD fields were taken from 1990s and 2090s averages of ACCESS1.3 simulations.) The unrealistically deep MLDs of the 1990s (used here as the preindustrial state) dramatically shoal in the 2090s, partially shutting off the model's deep-water formation (Bi et al., 2013) although the deep Atlantic MLD

near the Weddell Sea has not entirely subsided in the RCP4.5 scenario (Fig. C2b).

The dominant control on the sequestration strength of the biological pump is the mean time $\Gamma^{\uparrow}$ to return water to the euphotic layer (termed reexposure time; Primeau, 2005; DeVries and Holzer, 2019; Holzer et al., 2020), because it sets the mean residence (or sequestration) time of regenerated DIC. Figure C3 shows the Atlantic, Pacific, and Indian Ocean zonal mean $\Gamma^{\uparrow}$ for the preindustrial and future states. (Reexposure times are calculated as in the work by Holzer et al. (2020) who considered only the RCP8.5 scenario.) $\Gamma^{\uparrow}$ increases greatly (by up to 600 and 1200 years for the RCP4.5 and RCP8.5

scenarios, respectively). At first glance the RCP8.5 response may appear to be proportional to the RCP4.5 response, but on





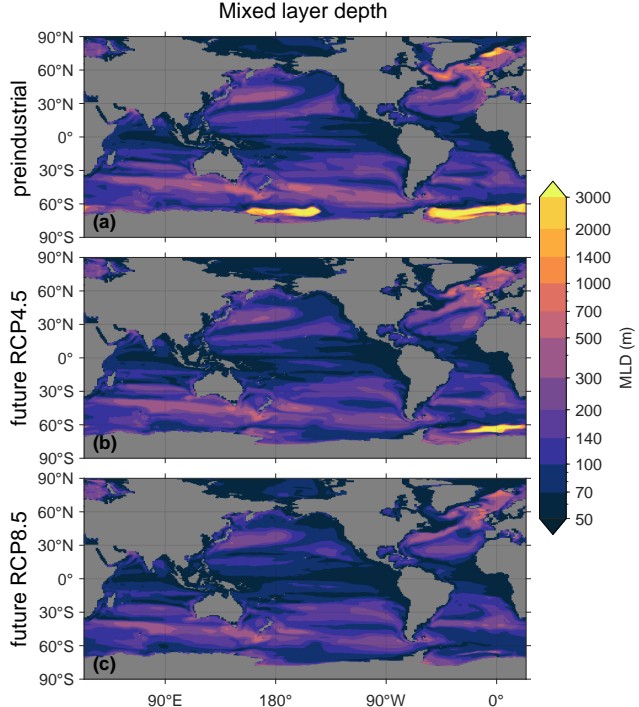

**Figure C2.** (a) Mixed layer depth (MLD) for the preindustrial state. (b) As (a) for the future RCP4.5 state. (c) As (b) but for RCP8.5.

closer inspection different patterns are visible, particularly in the Atlantic sector. These differences are consistent with the corresponding differences in MLD (Fig. C2).



**Figure C3.** (a)–(c) Mean water reexposure time, $\Gamma^\uparrow$, for the preindustrial circulation, zonally averaged over the Atlantic (a), Pacific (b), and Indian Ocean (c). (d)–(f) As (a)–(c) but for the RCP4.5 state. (g)–(i) As (d)–(f) but for RCP8.5. (j)–(o) As (d)–(i) but for $\Delta\Gamma^\uparrow$.



## Appendix D: Biological production and export production

Figure D1 shows the production, $U$, for the preindustrial and future states, along with the corresponding changes. Similarly, Fig. D2 shows the corresponding export ratios, $f$. As shown by Fig. 3, important contributions from export ratios occur in places where $\Delta f$ and $U$ are both large, and important contributions from production occur where $f$ and $\Delta U$ are both large.

The changes in export production vary with mechanism, i.e., $POC_f$, $POC_s$, DOC, or PIC. The zonally integrated changes for each mechanism are shown in Fig. D3. For $POC_s$ export, changes are strongly dominated by export ratios, particularly in the tropics likely because very little $POC_s$ makes it through the euphotic zone ($f \approx 0$) and thus $\Delta J_{ex} \approx U\,\Delta f$. In contrast, PIC export is entirely controlled by changes in production ($\Delta J_{ex} \approx f\,\Delta U$; Fig. D3) because in our model its export ratio is unaffected by any environmental change ($f \approx 0.97$ and $\Delta f \approx 0.001$).

*Author contributions.* BP and MH designed the study, analyzed the data, and wrote the manuscript with input from all co-authors. BP wrote the code, performed the experiments, and produced the figures with input from all co-authors. The funding was acquired by MH.

*Competing interests.* The authors declare that they have no conflict of interest.

*Acknowledgements.* We thank Yi Liu for helpful discussions. This work was supported by Australian Research Council grant DP210101650 (to MH) and undertaken with the assistance of resources and services from the National Computational Infrastructure (NCI), which is supported by the Australian Government.



**Figure D1.** As Fig. 2 but for DIC uptake, $U$ (biological production).





**Figure D2.** As Fig. 2 but for export ratios, $f$.



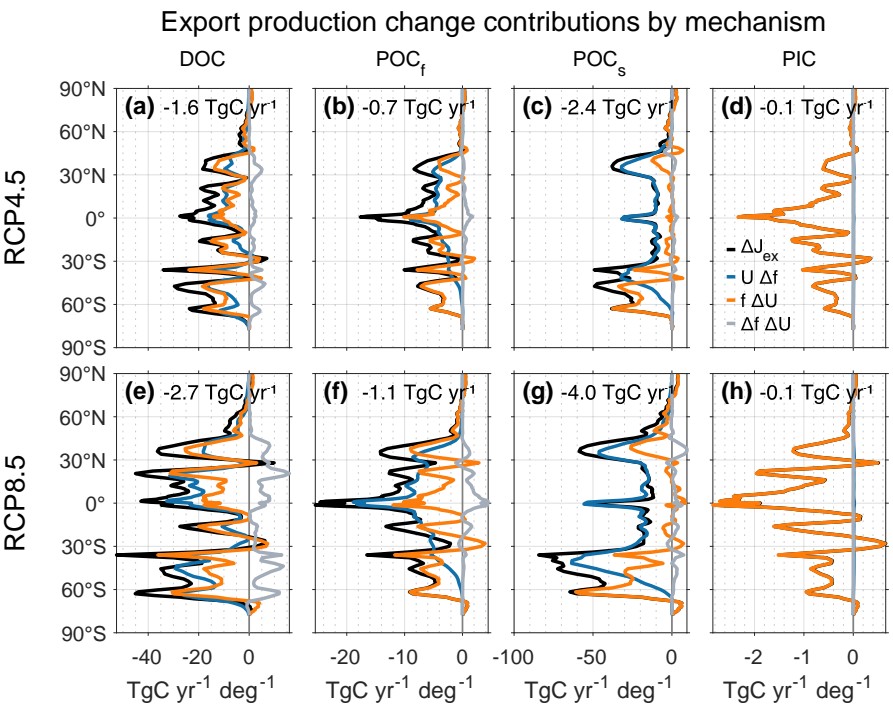

**Figure D3.** (a)–(d) Change in DOC (a), POC$_f$ (b), POC$_s$ (c), and PIC (d) zonally integrated export production, $\Delta J_{\mathrm{ex}}$ (black), for the RCP4.5-based state, decomposed into contributions from $U\,\Delta f$ (blue), $f\,\Delta U$ (orange), and $\Delta f\,\Delta U$ (gray). The globally integrated contribution to $\Delta J_{\mathrm{ex}}$ is indicated in each panel. (e)–(h) As (a)–(d) but for the RCP8.5-based state.



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
