# Peer review of "The ocean's biological and preformed carbon pumps in perpetually slower and warmer oceans"

_EGUsphere, 2023_

## Author Comment (AC1)

**Response to egusphere-2023-2525 RC1**

Below is our point-by-point response to RC1. Referee #1's comments are in *gray italic*. Our responses are in black. Expected manuscript revisions are indicated in blue.

*General comments:*

*The paper of Pasquier et al. investigates the future changes of the ocean's biological and preformed carbon pumps by 2090s under two emission scenarios (RCP4.5 and RCP8.5) using a simple model biogeochemistry model and steady-state ocean circulation transport matrix models. They present a new partitioning of preformed DIC to separate the contributions of different pathways of preformed DIC to the ocean carbon storage and outgassing. Using this data-constrained model approach, they found that biological production declines only modestly in the future, while organic matter export declines more significantly due to the reductions in both biological production and export ratio.*

*The paper is well-written with clear figures and presents interesting results on the biogeochemical cycling and biological carbon pump. I only have specific minor comments and questions detailed below:*

We thank Referee #1 for their positive general comments.

- *The authors highlight the fact that the model results are to some extent imprinted by unrealistic circulation features of the ACCESS1.3 model, especially in the Southern Ocean and the deep ocean. The sensitivity of the results to the choice of this peculiar model should be more emphasize.*

We agree that these features have important effects on the ocean carbon cycle, as we point out throughout the manuscript. Without direct comparison with other circulation models it is difficult to quantify the sensitivity. While deep convection that reaches to the bottom in the Southern Ocean is unrealistic, this turns out to be the ACCESS1.3 mechanism for forming AABW, and ACCESS1.3 reduces AABW formation in the future. Reduced AABW formation in the future is a feature that is robust in the sense that it is seen in a number of models and it is also supported by recent observations (de Lavergne et al., 2014). We therefore expect the large-scale changes we document not to be particularly sensitive to the fact that AABW is formed through unrealistic means. In response, we will add to the Discussion something along the following lines:

> While the ACCESS1.3 model has unrealistically deep convection in the Southern Ocean, this deep convection forms much of the model's AABW (Bi et al., 2013), and the future reduction in AABW seen here is robust across different models and also expected from recent observations (de Lavergne et al., 2014). We therefore do not expect our results to be particularly sensitive to this unrealistic model feature.

- *This study does not account for the potential changes in the oceanic circulation and stratification due to melting of ice sheets, which could alter the ventilation and storage of carbon in the deep ocean. If this were the case, would it result in different behavior of the biological production or organic matter export?*

Referee #1 is correct that ACCESS1.3 does not model meltwater input from terrestrial ice sheets (as is the case for most climate models, as far as we are aware). We agree that melting ice sheets will have significant effects on biological production and organic-matter export through major changes in the global circulation (Li et al., 2023; Purich et al., 2018). While we cannot give a definitive answer, we would expect the effect from ice-sheet meltwater to exacerbate the reduction in future Southern Ocean ventilation (this was the case for meltwater experiments using the ACCESS-ESM1.5 model in the SSP5-8.5 scenario; (Purich & England, 2023)). In response, we will add to the Discussion something along the lines of:

> We note that while the parent ACCESS model includes effects from sea-ice melting on the ocean circulation, it does not include melting terrestrial ice sheets. If meltwater from ice sheets were included, we would expect a stronger reduction in Southern Ocean ventilation (Purich et al., 2018, 2023; Li et al., 2023). This could in turn strengthen the already dominating role of the circulation in driving changes in the biological and preformed carbon pumps discussed here.

- *Why focusing on the RCP4.5 and RCP8.5 scenarios? This has to be justified.*

We chose the RCP8.5 scenario simply because, of the commonly studied and available future scenarios, it provides the strongest perturbation of the ocean state and hence a very clear signal for us to quantify. Adding the more likely intermediate RCP4.5 scenario allowed us to quantify sensitivity to future scenario by additionally studying a more realistic perturbation. In response, we plan to add the following to Section 2.1 (Ocean Circulation Models):

> We use the Representative Concentration Pathway RCP8.5 (which represents the worst-case scenario for future global warming; Meinshausen et al., 2011) because it provides the strongest perturbation, along with the more likely, intermediate RCP4.5 to assess the sensitivity of our results to climate-change scenario.

- *What are the implications of prescribed pCO2 concentrations for the simulation? It would have been interesting to account for the feedbacks between the ocean carbon cycle and the atmospheric pCO2 concentration, which may affect the future evolution of the preformed DIC.*

The implications for our simulations are that changing the prescribed atmospheric $p\mathrm{CO_2}$ has a negligible effect on everything except the preformed DIC pipes. While we agree that the feedbacks on atmospheric $p\mathrm{CO_2}$ are important (and has been the subject of previous work by the authors, e.g., Holzer et al. (2021)), they are out of the scope of our study.

If the total (i.e., ocean + atmosphere) carbon inventory were prescribed at its 2090s levels — instead of prescribing atmospheric $p\mathrm{CO_2}$ — then most of the carbon would ultimately end up in the ocean because of the buffer factor, resulting in an atmospheric $p\mathrm{CO_2}$ close to its preindustrial value. Furthermore, the prescribed atmospheric $p\mathrm{CO_2}$ does not affect anything other than the mean

preformed DIC concentration. For example, we found that prescribing $pCO_2 = 360$ µatm (the mean 1990s value, which differs from the preindustrial 270 µatm used here) for our preindustrial state merely shifts DIC concentrations up by the same amount at every location. This uniform shift in DIC would have little effect on our results because no process in the ocean interior depends on the global mean DIC concentrations. This is demonstrated by the following unpublished figure of global-mean tracer profiles (see the yellow and green lines, for which we used the preindustrial circulation but with different $pCO_2$ values):

[Figure]

Figure 1: Global mean depth profiles of (a) $PO_4$, (b) dissolved $O_2$, (c) DIC, and (d) total alkalinity for the preindustrial state (green and yellow lines) and future states (orange and red lines). Note how the green and yellow lines only differ for DIC (panel c).

In response, we will add something along these lines in the Discussion:

> Note that the changes in prescribed atmospheric $pCO_2$ dominate the changes in air–sea exchange and strongly affect the flowrates of preformed DIC through ingassing and degassing. Hence, the flowrates of the preformed DIC pipes in Fig. 6 would be significantly different if we had prescribed the total (i.e., oceanic plus atmospheric) carbon inventory instead of only the atmospheric carbon inventory.

- *I found the description of the preformed carbon pump and how it differs from the solubility pump a bit confusing. I think it could be clarified.*

We will clarify the distinction in the Introduction to read something similar to (addition in bold):

> To comprehensively track all carbon through the ocean, we quantify future change in terms of the biological carbon pump and in terms of what we call the preformed carbon pump. A quantification of the preformed pump is made possible by a novel partition of preformed DIC according to its sources and sinks. The preformed pump can be regarded as the solubility pump, although the latter term is often used specifically for the subduction of DIC driven by solubility gradients (e.g., Volk and Hoffert, 1985). **The preformed pump defined here allows us to track all preformed DIC back to its source when**

> **atmospheric CO2 entered the ocean, when organic matter was remineralized in the surface ocean, or when aphotically regenerated DIC resurfaced. In addition, we track preformed DIC forward to its sinks, when it will degas back to the atmosphere or be utilized biologically.** In this way we quantify the timescales and flow rates (the "plumbing") of not just the biological pump but also of the preformed pump, as well as the interaction between these pumps and the atmosphere and how these change in the future under idealized steady-state scenarios.

- *As explained by the authors, the effect of iron on phytoplankton growth is not included in this study to avoid complexity. How would it change the presented results if it was taken into account? (i.e. does it have an important role?)*

It is difficult to answer this question quantitatively without explicitly modelling the iron cycle, which is out of the scope here. However, our PCO2 model captures the global patterns of iron limitation in the late 20th-century ocean because the model was optimized against observed nutrient concentrations (Pasquier et al., 2023), which are partly controlled by the iron cycle. In addition, we expect that future changes in the global iron supply will not play a significant role in the carbon cycle (Tagliabue et al., 2014). Iron deposition and its effect on biological production are predicted to be order 10% by 2100 for RCP8.5 with strong spatial variations and compensations (Drenkard et al., 2023; Liu et al., 2024). For example, Liu et al. (2023) did not report any biological-pump effect from changes in the iron cycle (even though the CESMv1 model used includes iron). Thus, while the iron cycle could change dramatically on very long timescales, it is reasonable to assume that its effects would be dominated by the large effects from circulation changes as considered here in our steady-state analysis. No change expected.

*Other comments:*

*L6: "experiences only modest declines". Precise how much.*
We will add "8–12 %".

*L8: the latter being driven*
"being" will be added.

*L19: In recent decades,*
Comma will be added.

*L30: …pump, preindustrial atmospheric pCO2 concentrations*
Comma after "pump" will be added and "p" will be added to read "$p\text{CO}_2$".

*L55: remove the brackets of the sentence. Same at L77-78, L90-91, L128-129, L132, L258-259, L320-321, L375-376, L452-453 and in the caption of Figure 4.*
We agree that too many parentheses reduces readability. However, we think that some of our parenthetical sentences should remain parenthetical to make it easy for the reader to identify details that could be skipped on a first reading. We will defer to the Editor, but the changes we expect to apply are:
- L55: OK
- L77-78: OK
- L90-91: OK

- L128-129: no change
- L132: OK
- L258-259: OK
- L320-321: OK
- L375-376: no change
- L452-453: OK
- caption of Figure 4: OK

*L66-71: this paragraph has to be removed from the introduction since it is about the results of the study.*
We agree that this paragraph can be removed, and we will do so in the revisions. We would like to note, however, that this type of "upshot paragraph" is fairly commonplace and often gives a useful preview of essence of the paper. Whether to include an upshot paragraph or not is mostly a matter of style. In this instance, we agree that it is not necessary.

*L80: atmospheric pCO2. Same at L93.*
L80 is correct as we are talking about the $CO_2$ mixing ratio. No change expected.
L93 is also correct. However, as suggested by Referee #1, we will change "atmospheric $CO_2$ concentrations" to "atmospheric $p CO_2$" here.

*L105-106: precise the reference used to choose the parameters of the PCO2 model*
The choice of parameters in the PCO2 model was described in detail in our previous study (Pasquier et al., 2023). We will add the reference in the manuscript.

*L112: …which in steady state obeys:*
No punctuation here is not incorrect and a matter of house style, which we leave for the copy editor to decide. No change expected (unless required by copy editor).

*L133: the ocean through pCO2 air-sea exchange*
Partial pressures are not exchanged, $CO_2$ molecules are. No change expected

*L138: Note that in our model, carbon*
Comma will be added.

*L156: "the exact same concentrations as traditionally defined preformed DIC". This has to be precised. What are these concentrations? How are they chosen?*
Note that the traditionally-defined preformed concentrations are described in the previous sentence. For clarity, we will change to "as obtained from the traditional propagation approach".

*L178: what are the main mechanisms*
This sentence is intended as a signpost to guide the reader by indicating what comes next, including a description of these main mechanisms. No change expected.

*L183: remove merely*
OK.

*L186: "the more sluggish circulation". Precise the sentence with value for the circulation.*
There is no unique metric for circulation sluggishness. The point here is that biological production declines together with the nutrient supply, and that this decline is expected given that the circulation is slower in our future states than in the preindustrial. In this paragraph we report the nutrient decline in the euphotic zone and then discuss its drivers. Adding a metric of the circulation change here would distract from the main point. Such a metric would need to be explained and motivated, e.g., justified through its relation with mean euphotic nutrient concentrations, which is so much detail that it would derail our argument. Having said this, to give you an idea of how much the deep circulation slows down, the mean age and mean reexposure time change by up to 600 and

1200 years for the RCP4.5- and RCP8.5-based states, respectively (see Fig. C3 in this manuscript or the workd of Holzer et al. (2020)). No change expected.

*L186: decline by 12 and 19 % for the two scenarios compared to the preindustrial*
For readability, we avoided repeating "compared to the preindustrial state" every time when this is clear from context, which we think is the case here. The two-sentence lead paragraph of Section 3.1 includes "We now examine how each of these components change from their preindustrial values". No change expected.

*L187-188: "due to the slower future circulation and decreased ventilation". Please give values.*
As for first L186 point above, it would be distracting to dive into circulation/ventilation metrics at this stage of the manuscript. No change expected.

*L191: for RCP4.5 and RCP8.5 respectively (compared to the preindustrial)*
As for point above, we also avoided repeating "respectively" over and over, particularly when unambiguous from the context, as we think is the case here. No change expected.

*L191: "although the North Atlantic contains a patch of prominent cooling". Does it have a local effect on nutrient and carbon uptake rates?*
Yes, because the nutrient and carbon uptake rates are temperature dependent (Eq. (2)). However, this effect is not widespread enough to impact the global-scale response. No change expected.

*L204: RCP4.5 and RCP8.5 respectively*
Will add "respectively".

*L220-221: in the RCP4.5 and RCP8.5 scenarios respectively, compared to preindustrial. Same in L253.*
As for L186 point above, that the changes mentioned are relative to the preindustrial state is obvious from the context. No change expected L220–221. However, for L253, we will add that Table 1 also lists the "corresponding preindustrial-to-future changes" in the preceding sentence.

*L230: "do not suffice". Replace by are not enough.*
This is correct common-language English. No change expected.

*Table 1: Add to the caption "...for preindustrial, RCP4.5 and RCP8.5 scenarios. Variations relative to the preindustrial are also shown".*
OK.

*L271: remove a priori*
OK.

*L275: remove such*
OK.

*L302: and how they change in the future*
We will replace "these" with "they".

*L303: remove the [t]*
OK.

*Figure 6: make the dashed rectangle in the left corner more visible for the reader.*
OK.

*L326: To better understand the transport pathways of preformed DIC, we quantify the amount able to enter*
OK.

*L328: (not shown in Fig. 6) revealed that regardless of source-sink pair, more*
OK.

*L357: Italic font may be used for emphasis and used sparingly. Remove the italic style of the word less. Same in L361, 364, 365.*

The emphasis usage in this paragraph is important because it brings attention to the subtle and surprising differences between pump strength and pump efficiency as we quantify them here. Italic font is used for emphasis for 5 words out of roughly 15,000 for the whole article. No change expected.

*L437-439: remove the sentence to explain the SSP scenarios since it is not useful.*

We respectfully disagree that explaining how RCPs match SSPs is not useful, but we agree that this passage is too verbose. We will rephrase succinctly as "(which nominally match RCP4.5 and RCP8.5; Riahi et al., 2017, Arias et al., 2021)".

OK.

We think this is correct usage but will defer to the editor. The "Liu et al." citation without the year is done through the `\citeauthor` command in LaTeX, which is suggested in the publisher's LaTeX template:

```
%% LITERATURE CITATIONS
%%
%% command & example result
%% citet{jones90}| & Jones et al. (1990)
%% citep{jones90}| & (Jones et al., 1990)
%% citep{jones90,jones93}| & (Jones et al., 1990, 1993)
%% citep[p. 32]{jones90}| & (Jones et al., 1990, p. 32)
%% citep[e.g.,][]{jones90}| & (e.g., Jones et al., 1990)
%% citep[e.g.,][p. 32]{jones90}| & (e.g., Jones et al., 1990, p. 32)
%% citeauthor{jones90}| & Jones et al.
%% citeyear{jones90}| & 1990
```

This avoids unnecessarily repeating the year here, which is unambiguous given that Liu et al. (2023) was referenced at the start of the paragraph and that there is only one reference for Liu et al. No change expected unless other Liu et al. references are added.

This is appropriate English for what was done in this work. No change expected.

This is correct English. No change expected.

*L496: with declines of ~10 %, even for RCP8.5. Remove the italic style as well.*

Will remove "merely" and add the comma after "10 %". This paragraph contains no italic style.

*L500: RCP8.5-based steady-state scenarios respectively, in comparison to preindustrial*

Will add "respectively" here but will add "compared to the preindustrial" to the previous bullet point instead.

*References: sort the references of DeVries in chronological order.*

References are automatically sorted alphabetically by the publisher's LaTeX template. No change expected.

**References (including those already cited in the manuscript)**

Bi, D., Marsland, S. J., Uotila, P., O'Farrell, S., Fiedler, R. A. S., Sullivan, A., Griffies, S. M., Zhou, X., & Hirst, A. C. (2013). ACCESS-OM: The Ocean and Sea Ice Core of the ACCESS Coupled Model. *Australian Meteorological and Oceanographic Journal*, *63*, 213–232.

de Lavergne, C., Palter, J. B., Galbraith, E. D., Bernardello, R., & Marinov, I. (2014). Cessation of Deep Convection in the Open Southern Ocean under Anthropogenic Climate Change. *Nature Climate Change*, *4*(4), 278–282. https://doi.org/10.1038/nclimate2132

Drenkard, E. J., John, J. G., Stock, C. A., Lim, H.-G., Dunne, J. P., Ginoux, P., & Luo, J. Y. (2023). The Importance of Dynamic Iron Deposition in Projecting Climate Change Impacts on Pacific Ocean Biogeochemistry. *Geophysical Research Letters*, *50*(21), e2022GL102058. https://doi.org/10.1029/2022GL102058

Holzer, M., Chamberlain, M. A., & Matear, R. J. (2020). Climate-Driven Changes in the Ocean's Ventilation Pathways and Time Scales Diagnosed From Transport Matrices. *Journal of Geophysical Research: Oceans*, *125*(10), e2020JC016414. https://doi.org/10.1029/2020JC016414

Holzer, M., Kwon, E. Y., & Pasquier, B. (2021). A New Metric of the Biological Carbon Pump: Number of Pump Passages and Its Control on Atmospheric pCO\textsubscript{2}. *Global Biogeochemical Cycles*, *35*(6), e2020GB006863. https://doi.org/10.1029/2020GB006863

Li, Q., England, M. H., Hogg, A. M., Rintoul, S. R., & Morrison, A. K. (2023). Abyssal Ocean Overturning Slowdown and Warming Driven by Antarctic Meltwater. *Nature*, *615*(7954), 841–847. https://doi.org/10.1038/s41586-023-05762-w

Liu, J., Wang, X., Wu, D., Wei, H., Li, Y., & Ji, M. (2024). Historical Footprints and Future Projections of Global Dust Burden from Bias-Corrected CMIP6 Models. *Npj Climate and Atmospheric Science*, *7*(1), 1–12. https://doi.org/10.1038/s41612-023-00550-9

Liu, Y., Moore, J. K., Primeau, F., & Wang, W. L. (2023). Reduced CO2 Uptake and Growing Nutrient Sequestration from Slowing Overturning Circulation. *Nature Climate Change*, *13*(1), 83–90. https://doi.org/10.1038/s41558-022-01555-7

Pasquier, B., Holzer, M., Chamberlain, M. A., Matear, R. J., Bindoff, N. L., & Primeau, F. W. (2023). Optimal parameters for the ocean's nutrient, carbon, and oxygen cycles compensate for circulation biases but replumb the biological pump. *Biogeosciences*, *20*(14), 2985–3009. https://doi.org/10.5194/bg-20-2985-2023

Purich, A., & England, M. H. (2023). Projected Impacts of Antarctic Meltwater Anomalies over the Twenty-First Century. *Journal of Climate*, *36*(8), 2703–2719. https://doi.org/10.1175/JCLI-D-22-0457.1

Purich, A., England, M. H., Cai, W., Sullivan, A., & Durack, P. J. (2018). Impacts of Broad-Scale Surface Freshening of the Southern Ocean in a Coupled Climate Model. *Journal of Climate*, *31*(7), 2613–2632. https://doi.org/10.1175/JCLI-D-17-0092.1

Tagliabue, A., Aumont, O., & Bopp, L. (2014). The Impact of Different External Sources of Iron on the Global Carbon Cycle. *Geophysical Research Letters*, *41*(3), 920–926. https://doi.org/10.1002/2013GL059059

---

## Author Response (AR1)

**Response to Referee Comments**
**egusphere-2023-2525**

Below is our response to Referee Comments RC1 and RC2. This document combines the author comments AC1 and AC2 that have been previously submitted during the initial discussion phase, but updated to reflect the submitted revisions of the manuscript. Referee comments are in *gray italic*. Our responses are in black. Manuscript revisions are indicated in blue.

Note that in addition to the revisions in response to RC1 and RC2, we have carefully read through the manuscript to correct typos and we have slightly improved the quality of the text and figures.

**Contents**

**Response to RC1**

(updated from AC1 to reflect the revised manuscript submission.)

*General comments:*

*The paper of Pasquier et al. investigates the future changes of the ocean's biological and preformed carbon pumps by 2090s under two emission scenarios (RCP4.5 and RCP8.5) using a simple model biogeochemistry model and steady-state ocean circulation transport matrix models. They present a new partitioning of preformed DIC to separate the contributions of different pathways of preformed DIC to the ocean carbon storage and outgassing. Using this data-constrained model approach, they found that biological production declines only modestly in the future, while organic matter export declines more significantly due to the reductions in both biological production and export ratio.*

*The paper is well-written with clear figures and presents interesting results on the biogeochemical cycling and biological carbon pump. I only have specific minor comments and questions detailed below:*

We thank Referee #1 for their positive general comments.

- *The authors highlight the fact that the model results are to some extent imprinted by unrealistic circulation features of the ACCESS1.3 model, especially in the Southern Ocean and the deep ocean. The sensitivity of the results to the choice of this peculiar model should be more emphasize.*

We agree that these features have important effects on the ocean carbon cycle, as we point out throughout the manuscript. Without direct comparison with other circulation models it is difficult to quantify the sensitivity. While deep convection that reaches to the bottom in the Southern Ocean is unrealistic, this turns out to be the ACCESS1.3 mechanism for forming AABW, and ACCESS1.3 reduces AABW formation in the future. Reduced AABW formation in the future is a feature that is robust in the sense that it is seen in a number of models and it is also supported by recent observations (de Lavergne et al., 2014). We therefore expect the large-scale changes we document not to be particularly sensitive to the fact that AABW is formed through unrealistic means. In response, we have added the following to the Discussion section:

> iii. Most quantitative aspects of our results are likely model specific and to some extent imprinted by circulation biases. While the ACCESS1.3 parent model is state-of-

> the-art, it produces Antarctic Bottom Water (AABW) through unrealistically deep convection in the Southern Ocean (Bi et al., 2013). However, the future reduction in AABW seen here is robust across CMIP5 models and also expected from recent observations (de Lavergne et al., 2014). We therefore do not expect our results to be sensitive to this unrealistic model feature. Although a model with more realistic AABW formation might produce quantitatively different results (e.g., production could increase due to a smaller decline in mixed layer depths), the qualitative links between changes and their driving mechanisms should be robust and model-independent (e.g., mixed-layer shoaling driving intensified nutrient trapping; Liu et al., 2023).

- *This study does not account for the potential changes in the oceanic circulation and stratification due to melting of ice sheets, which could alter the ventilation and storage of carbon in the deep ocean. If this were the case, would it result in different behavior of the biological production or organic matter export?*

Referee #1 is correct that ACCESS1.3 does not model meltwater input from terrestrial ice sheets (as is the case for most climate models, as far as we are aware). We agree that melting ice sheets will have significant effects on biological production and organic-matter export through major changes in the global circulation (Li et al., 2023; Purich et al., 2018). While we cannot give a definitive answer, we would expect the effect from ice-sheet meltwater to exacerbate the reduction in future Southern Ocean ventilation (this was the case for meltwater experiments using the ACCESS-ESM1.5 model in the SSP5-8.5 scenario; (Purich & England, 2023)). In response, we have added the following to the Discussion section:

> iv. The parent ACCESS1.3 model does not include meltwater from terrestrial ice sheets. Including ice-sheet meltwater would further reduce Southern Ocean ventilation (Li et al., 2023; Purich et al., 2018; Purich & England, 2023). This would likely strengthen the already dominating role of the circulation in driving changes in the biological and preformed carbon pumps.

- *Why focusing on the RCP4.5 and RCP8.5 scenarios? This has to be justified.*

We chose the RCP8.5 scenario simply because, of the commonly studied and available future scenarios, it provides the strongest perturbation of the ocean state and hence a very clear signal for us to quantify. Adding the more likely intermediate RCP4.5 scenario allowed us to quantify sensitivity to future scenario by additionally studying a more realistic perturbation. In response, we have added the following to Section 2.1 (Ocean Circulation Models):

> For the future ocean states, we use the 2090s average for the ACCESS1.3 CMIP5 runs for the RCP4.5 and RCP8.5 scenarios. We use RCP8.5 (which represents the worst-case scenario for future global warming; Meinshausen et al., 2011) because it provides the strongest perturbation, and the more likely intermediate RCP4.5 to assess the sensitivity of our results to climate-change scenario. We prescribe atmospheric $CO_2$ ratios at 278, 536, and 886 ppm for the preindustrial-, RCP4.5-, and RCP8.5-based states analyzed below.

- *What are the implications of prescribed pCO2 concentrations for the simulation? It would have been interesting to account for the feedbacks between the ocean carbon cycle and the atmospheric pCO2 concentration, which may affect the future evolution of the preformed DIC.*

The implications for our simulations are that changing the prescribed atmospheric $p\mathrm{CO_2}$ has a negligible effect on everything except the preformed DIC pipes. While we agree that the feedbacks on atmospheric $p\mathrm{CO_2}$ are important (and has been the subject of previous work by the authors, e.g., Holzer et al. (2021)), they are out of the scope of our study.

If the total (i.e., ocean + atmosphere) carbon inventory were prescribed at its 2090s levels — instead of prescribing atmospheric $p\mathrm{CO_2}$ — then most of the carbon would ultimately end up in the ocean because of the buffer factor, resulting in an atmospheric $p\mathrm{CO_2}$ close to its preindustrial value. Furthermore, the prescribed atmospheric $p\mathrm{CO_2}$ does not affect anything other than the mean preformed DIC concentration. For example, we found that prescribing $p\mathrm{CO_2}$ = 360 µatm (the mean 1990s value, which differs from the preindustrial 270 µatm used here) for our preindustrial state merely shifts DIC concentrations up by the same amount at every location. This uniform shift in DIC would have little effect on our results because no process in the ocean interior depends on the global mean DIC concentrations. This is demonstrated by the following unpublished figure of global-mean tracer profiles (see the yellow and green lines, for which we used the preindustrial circulation but with different $p\mathrm{CO_2}$ values):

[Figure]

Figure 1: Global mean depth profiles of (a) $PO_4$, (b) dissolved $O_2$, (c) DIC, and (d) total alkalinity for the preindustrial state (green and yellow lines) and future states (orange and red lines). Note how the green and yellow lines only differ for DIC (panel c).

In response, we have added the following to the Section 3.3.2 (The future preformed pump) when discussing Fig. 6:

> Across all source–sink pairs, the ingassing-to-outgassing flow rates increase the most (roughly by factors 3 and 6 for RCP4.5 and RCP8.5) because of the dramatic change in atmospheric $p\mathrm{CO_2}$. Note that, had we prescribed the total (i.e., oceanic plus atmospheric) carbon inventory instead of only the atmospheric $p\mathrm{CO_2}$, the increase in atmospheric $p\mathrm{CO_2}$ and in the ingassing/outgassing fluxes in Fig. 6 would have been significantly smaller.

- *I found the description of the preformed carbon pump and how it differs from the solubility pump a bit confusing. I think it could be clarified.*

We have clarified the distinction in two places. In the Introduction, the corresponding passage now reads:

> To comprehensively track all carbon through the ocean, we quantify future change in terms of the biological carbon pump and in terms of what we call the preformed carbon pump. A quantification of the preformed pump is made possible by a novel partition of preformed DIC according to its sources and sinks. The preformed pump

> defined here allows us to track all preformed DIC back in time to its sources when atmospheric $CO_2$ entered the ocean, when organic matter was remineralized in the euphotic zone, or when aphotically regenerated DIC resurfaced. In addition, we track preformed DIC forward in time to its sinks, when it will outgas to the atmosphere or be utilized biologically. In this way we quantify the timescales and flow rates (the plumbing'') of not just the biological pump but also of the preformed pump, as well as the interaction between these pumps and the atmosphere and how these change in the future under idealized steady-state scenarios.

In addition, Section 3.3 (The preformed carbon pump) now starts with:

> To assess the future state of the biological pump, it is useful to place it in the context of overall carbon sequestration by also considering the preformed DIC pool (e.g., Ito et al., 2015). Our approach differs conceptually from previous quantifications of the abiotic carbon cycle in terms of the solubility pump (e.g., Volk & Hoffert, 1985), where one typically focuses on the sub-surface pathways of preformed DIC, and efficiency is defined realtive to complete surface saturation. By contrast, our new preformed DIC tracer allows us to quantify not only interior but also euphotic DIC pathways, how they change in the future, and how they affect the ocean's entire DIC inventory. The "preformed carbon pump" considered here is thus the natural counterpart of the biological pump.

- *As explained by the authors, the effect of iron on phytoplankton growth is not included in this study to avoid complexity. How would it change the presented results if it was taken into account? (i.e. does it have an important role?)*

It is difficult to answer this question quantitatively without explicitly modelling the iron cycle, which is out of the scope here. However, our PCO2 model captures the global patterns of iron limitation in the late 20th-century ocean because the model was optimized against observed nutrient concentrations (Pasquier et al., 2023), which are partly controlled by the iron cycle. In addition, we expect that future changes in the global iron supply will not play a significant role in the carbon cycle (Tagliabue et al., 2014). Iron deposition and its effect on biological production are predicted to be order 10% by 2100 for RCP8.5 with strong spatial variations and compensations (Drenkard et al., 2023; Liu et al., 2024). For example, Liu et al. (2023) did not report any biological-pump effect from changes in the iron cycle (even though the CESMv1 model used includes iron). Thus, while the iron cycle could change dramatically on very long timescales, it is reasonable to assume that its effects would be dominated by the large effects from circulation changes as considered here in our steady-state analysis. No change.

*Other comments:*

*L6: "experiences only modest declines". Precise how much.*
We have added "8–12 %".

*L8: the latter being driven*
"being" has been added.

*L19: In recent decades,*
Comma has been added.

*L30: ...pump, preindustrial atmospheric pCO2 concentrations*
Comma after "pump" has been added and italic "*p*" has been added to read "$pCO_2$".

*L55: remove the brackets of the sentence. Same at L77-78, L90-91, L128-129, L132, L258-259, L320-321, L375-376, L452-453 and in the caption of Figure 4.*

We agree that too many parentheses reduces readability. However, we think that some of our parenthetical sentences should remain parenthetical to make it easy for the reader to identify details that could be skipped on a first reading. We will defer to the Editor, but the changes we have applied are:

- L55: no change
- L77-78: OK
- L90-91: OK
- L128-129: no change
- L132: OK
- L258-259: OK
- L320-321: OK
- L375-376: no change
- L452-453: OK
- caption of Figure 4: OK

*L66-71: this paragraph has to be removed from the introduction since it is about the results of the study.*

We have removed this paragraph. We would like to note, however, that this type of "upshot paragraph" is fairly commonplace and often gives a useful preview of essence of the paper. Whether to include an upshot paragraph or not is mostly a matter of style. In this instance, we agree that it is not necessary.

*L80: atmospheric pCO2. Same at L93.*

L80 is correct as we are talking about the $CO_2$ mixing ratio. No change expected.

L93 is also correct. However, as suggested by Referee #1, we have changed "atmospheric $CO_2$ concentrations" to "atmospheric $pCO_2$" here.

*L105-106: precise the reference used to choose the parameters of the PCO2 model*

The choice of parameters in the PCO2 model was described in detail in our previous study (Pasquier et al., 2023). We have added the reference.

*L112: ...which in steady state obeys:*

No punctuation here is not incorrect and a matter of house style, which we leave for the copy editor to decide. No change expected (unless required by copy editor).

*L133: the ocean through pCO2 air-sea exchange*

Partial pressures are not exchanged, $CO_2$ molecules are. No change.

*L138: Note that in our model, carbon*

Comma has been added.

*L156: "the exact same concentrations as traditionally defined preformed DIC". This has to be precised. What are these concentrations? How are they chosen?*

Note that the traditionally-defined preformed concentrations are described in the previous sentence. For clarity, we have changed to "as obtained by the traditional approach (i.e., by propagating euphotic concentrations)".

*L178: what are the main mechanisms*

This sentence is intended as a signpost to guide the reader by indicating what comes next, including a description of these main mechanisms. No change.

*L183: remove merely*

Done.

*L186: "the more sluggish circulation". Precise the sentence with value for the circulation.*
There is no unique metric for circulation sluggishness. The point here is that biological production declines together with the nutrient supply, and that this decline is expected given that the circulation is slower in our future states than in the preindustrial. In this paragraph we report the nutrient decline in the euphotic zone and then discuss its drivers. Adding a metric of the circulation change here would distract from the main point. Such a metric would need to be explained and motivated, e.g., justified through its relation with mean euphotic nutrient concentrations, which is so much detail that it would derail our argument. Having said this, to give you an idea of how much the deep circulation slows down, the mean age and mean reexposure time change by up to 600 and 1200 years for the RCP4.5- and RCP8.5-based states, respectively (see Fig. C3 in this manuscript or the work of Holzer et al. (2020)). No change.

*L186: decline by 12 and 19 % for the two scenarios compared to the preindustrial*
For readability, we avoided repeating "compared to the preindustrial state" every time when this is clear from context, which we think is the case here. The two-sentence lead paragraph of Section 3.1 includes "We now examine how each of these components change from their preindustrial values". No change.

*L187-188: "due to the slower future circulation and decreased ventilation". Please give values.*
As for first L186 point above, it would be distracting to dive into circulation/ventilation metrics at this stage of the manuscript. No change.

*L191: for RCP4.5 and RCP8.5 respectively (compared to the preindustrial)*
As for point above, we also avoided repeating "respectively" over and over, particularly when unambiguous from the context, as we think is the case here. No change.

*L191: "although the North Atlantic contains a patch of prominent cooling". Does it have a local effect on nutrient and carbon uptake rates?*
Yes, because the nutrient and carbon uptake rates are temperature dependent (Eq. (2)). However, this effect is not widespread enough to impact the global-scale response. No change.

*L204: RCP4.5 and RCP8.5 respectively*
Will have added "respectively".

*L220-221: in the RCP4.5 and RCP8.5 scenarios respectively, compared to preindustrial. Same in L253.*
As for L186 point above, that the changes mentioned are relative to the preindustrial state is obvious from the context. No change L220–221. However, for L253, we have added that Table 1 also lists the "corresponding preindustrial-to-future changes" in the preceding sentence.

*L230: "do not suffice". Replace by are not enough.*
This is correct common-language English. No change.

*Table 1: Add to the caption "...for preindustrial, RCP4.5 and RCP8.5 scenarios. Variations relative to the preindustrial are also shown".*
We have added:

> Total, regenerated, and preformed DIC inventories for the preindustrial, RCP4.5, and RCP8.5 states in units of PgC. Changes relative to the preindustrial state are also shown.

*L271: remove a priori*
Done.

*L275: remove such*
Done.

*L302: and how they change in the future*
We have replaced "these" with "they".

*L303: remove the [t]*
Sorry about this. Done.

*Figure 6: make the dashed rectangle in the left corner more visible for the reader.*
Done.

*L326: To better understand the transport pathways of preformed DIC, we quantify the amount able to enter*
Done.

*L328: (not shown in Fig. 6) revealed that regardless of source-sink pair, more*
We have added "in Fig. 6".

*L357: Italic font may be used for emphasis and used sparingly. Remove the italic style of the word less. Same in L361, 364, 365.*
The emphasis usage in this paragraph is important because it brings attention to the subtle and surprising differences between pump strength and pump efficiency as we quantify them here. Italic font is used for emphasis for 5 words out of roughly 15,000 for the whole article. No change.

*L437-439: remove the sentence to explain the SSP scenarios since it is not useful.*
We respectfully disagree that explaining how RCPs match SSPs is not useful, but we agree that this passage is too verbose. We have rephrased as:

> For CMIP6 models under the SSP2-4.5 and SSP5-8.5 scenarios (which nominally match RCP4.5 and RCP8.5; Riahi et al., 2017, Arias et al., 2021), the increase in ocean carbon sequestration is about 400–500 PgC by the year 2100 (Liu et al., 2023), of which only about 100 PgC is regenerated DIC as there has been insufficient time for it to accumulate at depth.

*L461: remove below*
Done.

*L462: Liu et al. (2023)*
We think this is correct usage but will defer to the editor. The "Liu et al." citation without the year is done through the `\citeauthor` command in LaTeX, which is suggested in the publisher's LaTeX template:

```
%% LITERATURE CITATIONS
%%
%% command & example result
%% citet{jones90}| & Jones et al. (1990)
%% citep{jones90}| & (Jones et al., 1990)
%% citep{jones90,jones93}| & (Jones et al., 1990, 1993)
%% citep[p. 32]{jones90}| & (Jones et al., 1990, p. 32)
%% citep[e.g.,][]{jones90}| & (e.g., Jones et al., 1990)
%% citep[e.g.,][p. 32]{jones90}| & (e.g., Jones et al., 1990, p. 32)
%% citeauthor{jones90}| & Jones et al.
%% citeyear{jones90}| & 1990
```

This avoids unnecessarily repeating the year here, which is unambiguous given that Liu et al. (2023) was referenced at the start of the paragraph and that there is only one reference for Liu et al. No change.

This is appropriate English for what was done in this work. No change.

*L491: this allows for the first time*
This is correct English. No change.

*L496: with declines of ~10 %, even for RCP8.5. Remove the italic style as well.*
We have removed "merely" and added the comma after "10 %". This paragraph contains no italic style.

*L500: RCP8.5-based steady-state scenarios respectively, in comparison to preindustrial*
Will have added "respectively" here but have added "from preindustrial values" to the previous bullet point instead.

*References: sort the references of DeVries in chronological order.*
References are automatically sorted alphabetically by the publisher's LaTeX template. No change.

**Response to RC2**

(updated from AC2 to reflect the revised manuscript submission.)

*Summary*

*The manuscript provides an analysis of the changes in the ocean carbon cycle between the preindustrial years and the 2090s, under the RCP4.5 and RCP8.5 future scenarios. Using a biogeochemical model forced by decadal-mean ocean circulations derived from a climate model, in steady-state, the authors find that (i) the counteracting effects of the decline in nutrient supply into the euphotic zone and the warming-enhanced phytoplankton growth rate induce a slight decrease in biological production, and (ii) the reduction in biological production and export ratio leads to a decrease in organic matter export. In addition, they assess the changes in preformed and regenerated dissolved inorganic carbon (DIC) inventories and cycles and find that due to the weakening of the circulation the inventory of regenerated and preformed DIC increases while the cycle of preformed DIC cycle becomes faster.*

*The manuscript makes a novel contribution by developing a partitioning of preformed and regenerated DIC in accounting for the source and sink processes. The manuscript is well written and organized. However I would like to raise several points, mostly regarding the discussion of the results, that should be addressed before its publication.*

*Please see the detailed explanation of these major points and all my detailed comments below.*

We thank Referee #2 for the positive feedback.

Before we provide detailed point-by-point responses, we would like to address Referee #2's frequent requests for uncertainty estimation (e.g., with repect to circulation choice, mixed-layer depth, averaging period, neglect of seasonality). While we share the referee's appreciation for the importance of uncertainty in general, we think that for our idealized study it makes little sense to estimate uncertainties even if that were feasible within the scope of our study. We are not making specific predictions for the future of the real ocean for which uncertainties would be important. Instead, our focus is conceptual on the processes that drive future changes in the marine carbon cycle for idealized steady-state scenarios. Importantly, the uncertainties that the referee would like to see are very unlikely to affect our conclusions.
Uncertainties with respect to model specifics could in principle be quantified by comparing a wide suite of different circulation and biogeochemical models, but that would necessitate major new research and is beyond the scope of our study. In previous work, Pasquier et al. (2023) had quantified the effects from the ACCESS-M circulation biases by embedding PCO2 in OCIM2, the ocean circulation inverse model (DeVries & Holzer, 2019). Pasquier et al. (2023) had

also quantified the effects from biogeochemical parameterization by altering its complexity or by changing biogeochemical parameter values. In the submitted manuscript, we focus on the conceptual and qualitative implications of our analysis without distracting the reader with uncertainties that are of little value in the context of our idealized study.

Below are our point-by-point responses.

*Main comments*

1. ***The circulation model****: The authors forced the biogeochemical model using the fields of the ACCESS-1.3 model. They referred to previous studies (Bi et al., 2020; Pasquier et al., 2023) stating a surestimation in the mixed layer depth in this model (in particular unrealistically deep mixed layer in the Weddell and Ross seas), that could impact the carbon pump. Several questions arise from the existence of these biases in the circulation model:*

- *Could the author justify the choice of the ocean circulation ACCESS-1.3 simulations for this study?*

We used ACCESS1.3 because it is a state-of-the art climate model and because this work builds on previous publications based on the ACCESS1.3 circulations. While we could have chosen a different circulation model, all models come with biases, and none of the other models' transport matrices have been assessed in as much detail as those for ACCESS1.3 (Chamberlain et al., 2019; Holzer et al., 2020). Furthermore, Pasquier et al. (2023) optimized the PCO2 biogeochemistry model embedded in the ACCESS1.3 circulation ("ACCESS-M PCO2") to quantify in detail the effects of circulation biases on the biological pump. Thanks to its optimized biogeochemical parameters, the ACCESS-M PCO2 model fits observations of DIC, total alkalinity, dissolved $O_2$, and $PO_4$ better than most CMIP5 and CMIP6 models (Bao & Li, 2016; Fu et al., 2022; Planchat et al., 2023). These factors make the ACCESS-M PCO2 model a natural choice for our study. In response, we have added the following paragraph after the caveats list in the Discussion section:

> A number of factors made the ACCESS-embedded PCO2 model a natural choice for this study, which builds on previous publications. While we could have chosen a different circulation model, all models have biases, and none of the other models' transport matrices have been assessed in as much detail as those for ACCESS1.3 (Chamberlain et al., 2019; Holzer et al., 2020). Furthermore, Pasquier et al. (2023) optimized PCO2 embedded in the ACCESS1.3 transport matrix ("ACCESS-M PCO2") and quantified the effects of circulation biases on the biological pump. Importantly, our optimized preindustrial state fits observations of DIC, total alkalinity, dissolved $O_2$, and $PO_4$ better than most CMIP5 and CMIP6 models (see, e.g., Bao & Li, 2016; Fu et al., 2022; Planchat et al., 2023).

- *The authors used optimized biogeochemical parameters, which partly corrects biases from circulation in the preindustrial conditions, if I correctly understand. Are these optimized parameters also used in the future scenario runs? If it is the case, how does this optimization influence carbon pump in the future states where these biases could have disappeared?*

Referee #2 is correct that the biogeochemical parameters were optimized for the preindustrial state and then also used for the future states. Optimizing the parameters for the future states (if future data were available) could fundamentally change the biogeochemistry, capturing, e.g., biological and ecosystem evolution, which is out of scope. Instead, the question here is how the carbon cycle changes due to changes in the physical state of the ocean at "fixed" biology (as modelled by PCO2), given that future biology is unknown. Importantly, we expect that our results are qualitatively robust to variations in parameter values because the mechanisms that

drive these changes operate in the same way regardless of the precise parameter values. In response, we have added the following sentence to Methods section 2.2 (Biogeochemistry Model):

> We use the same biogeochemical parameters for both preindustrial and future states to capture the response of the carbon cycle to changes in physical ocean state without any changes in plankton physiology.

- *Could the authors expand the third point in the discussion section L. 406-420 by specifying how these MLD biases impact their results on carbon flow rates and inventories, and their main conclusions, given the "key importance of circulation changes"? Could they estimate the uncertainties of the results related to these biases? If these uncertainties are not negligible, at least, the authors should qualify, in the abstract and conclusion sections, the quantified changes found between the preindustrial and 2090s in both future scenarios.*

One could estimate this uncertainty by applying our analysis to a family of different circulation models, but this is out of scope. While such uncertainties would be important when making predictions, our goal is different here: We used an idealized framework of steady future ocean states to elucidate the driving mechanisms causing changes in the ocean's carbon pumps. What is important for that goal is that the circulation model qualitatively captures the expected future changes (e.g., weakening of ventilation, increased stratification, slowing down of meridional overturning, weakening of deep-water formation, and so on). In response, we have qualified the abstract and conclusions accordingly. The abstract now contains the following sentence:

> We find that biological production experiences only modest declines (by 8–12 %) because the reduced nutrient supply due to a more sluggish future circulation and strongly shoaled mixed layers is counteracted by warming-stimulated growth.

The conclusion now contains the following sentence:

> The overall nutrient supply declines because of intensified Southern Ocean nutrient trapping and because decreased ventilation and strong mixed-layer shoaling slows the resurfacing of nutrients from depth.

2. ***The preindustrial conditions***: *For the preindustrial run, the authors forced the biogeochemical model using averages of circulation and thermodynamics over the 1990s instead of preindustrial years. They justified this choice based on the minor changes in hydrodynamics between these two periods. Why did they not directly use averages of circulation over preindustrial years? I suggest adding some explanations on this point in the Methods section.*

While we could have used a preindustrial mean, we chose the same period (1990s) as in previous work (Chamberlain et al., 2019; Holzer et al., 2020; Pasquier et al., 2023). The late-20th-century observations used to constrain the PCO2 parameters through optimization are more consistent with the 1990s circulation than with the preindustrial circulation. Regardless, we would expect preindustrial-to-1990s changes to be dwarfed by the 1990s-to-2090s changes, the latter being our focus here. We have added the following to Methods Section 2.1 (Ocean Circulation Models):

> To build our circulation models, we use physical ocean states from ACCESS1.3 climate-model simulations (Bi et al., 2013) for the preindustrial ocean and for two future climate scenarios. For the preindustrial ocean, we build on the work of Chamberlain et al. (2019) and Pasquier et al. (2023) and use the 1990s average of the circulation, thermodynamic, and forcing fields from the "historical" ACCESS1.3 runs submitted to CMIP5 (Taylor et al., 2012). We refer to this 1990s state as "preindustrial" because the carbon cycle was optimized against DIC observations corrected for anthropogenic DIC. The circulation itself and the other tracers used as constraints are appropriate for the 1990s, but because the preindustrial-to-1990s changes are dwarfed by the centennial changes analysed here, we do not make a further distinction with a true preindustrial state.

3. *Time integration: The authors considered averages on 10-year time slices. I suggest adding a small discussion on this relatively short time integration for which decadal variability is not handled.*

We of course agree that we subsample decadal variability with a 10-yr average, but we expect decadal variability to be small compared to the centennial-scale changes considered here. More importantly, our goal here is not to make precise predictions of carbon-cycle changes for which uncertainty due to decadal variability is important. Instead, our goal is to use a reasonable future state to explore the consequences of expected circulation changes on the carbon cycle. In response, we have added the following to the caveats list in Section 4 (Discussion):

> vi. Using the 1990s and 2090s decadal means from a single climate-model run does not capture decadal variability. However, we expect decadal variability to be small compared to the centennial changes considered here. While important for precise predictions, decadal variability is not important in our idealized steady-state framework for which any reasonable future state suffices.

4. *Seasonality: In the discussion section (L. 408-410), the authors also mentioned uncertainties in the results associated with the absence of seasonality in their circulation model forcing. Could the authors be more specific and give an estimate of these uncertainties in the carbon pump and its plumbing, associated with this simplification, or refer to previous studies that could have estimated them?*

While we cannot quantify the effect of seasonality with a steady-state circulation model, we expect seasonality to mostly affect the upper few hundred meters of the water column. Huang et al. (2021) recently built a seasonally varying ocean circulation inverse model (CYCLOCIM) and found that the inclusion of seasonality only improved the model–observations mismatch above roughly 200 m depth in the global mean. We therefore do not expect the absence of explicit seasonality to affect the qualitative character of our results, as most of the circulation-driven changes analyzed occur at depth. In response, we have added the following to the seasonality caveat in Section 4 (Discussion):

> ii. Our steady circulations do not capture seasonal covariances between physical, thermodynamic, and biological variations Riebesell et al., 2009). While this could be addressed with a cyclo-stationary model (e.g., Bardin et al., 2014), doing so would greatly increase complexity and computational cost. We expect seasonality to mostly affect the upper few hundred meters of the water column as was the case for Huang et al. (2021) who built a seasonally-varying ocean circulation inverse model (CYCLOCIM). We therefore do not expect the absence of explicit seasonality to affect the qualitative character of our results, as most of the circulation-driven changes analyzed occur at depth.

5. *Export of particulate organic matter: Pasquier et al. (2023) indicated that "particles are only submitted to gravitational sinking" in the coupled model. Is it also the case in the*

*present study? I suggest clarifying how the export of organic matter is calculated in the Methods section. If the transport of POC is similar as done in Pasquier et al. (2023), could the authors specify or give an estimate of how including the advective-diffusive transport could quantitatively change preformed and regenerated DIC inventories and carbon flow rates, in particular in deep convection areas?*

We use the same PCO2 as in the work of Pasquier et al. (2023), and POC is not transported by the circulation. This approximation is desirable for computational efficiency and justified because the gravitational transport of particles typically dominates the advective–diffusive transport (water currents are typically orders of magnitude smaller than our POC sinking velocities). Our goal was not to build the most realistic biogeochemistry model, but a relatively simple one with reasonable fidelity to observations in the current state of the ocean. (Important processes such as, e.g., temperature- and oxygen-dependent remineralization are parameterized in PCO2.) Furthermore, from (unpublished) numerical experiments, we do not expect that including the advective–diffusive transport of particles would make a significant difference to the model solutions. In response, we have added the following in Section 2.2 (Biogeochemistry Model):

> Organic matter is then remineralized back to DIC through respiration at rates $R_{\mathrm{DOC}}$, $R_{\mathrm{POCf}}$, $R_{\mathrm{POCs}}$, while PIC is dissolved to DIC at rate $D_{\mathrm{PIC}}$. In PCO2, biogenic particles are only transported by gravitational settling, which dominates advective–diffusive transport. All the particles that reach the bottom-most grid box are either remineralized or dissolved there.

*Detailed comments*

*L. 66-71: I suggest removing this text summarizing the main results of the study from the introduction section.*
Done.

*L. 113: Did the authors consider the deposition of particulate organic matter (POM) on the floor, remineralisation of POM in the sediment, and the flux of dissolved inorganic matter from the sediment to the water column?*
In PCO2, all POM that reaches the bottom is eventually remineralized in the grid box adjacent to the seafloor (Pasquier et al., 2023). This highly simplified benthic parameterization of sediment fluxes thus does not allow us to capture changes in benthic fluxes and early diagenesis that could be driven by global warming and changes in circulation. A more detailed parameterization of sediment fluxes could be implemented in future versions of PCO2, but is out of the scope here. In response, we have added that all the POC that reaches the seafloor is remineralized in Section 2.2 (Biogeochemistry Model; see response above) and have added the following to the caveats:

> v. Our model may not capture potentially important effects from biogeochemical mechanisms that are not explicitly parameterized (e.g., Henson et al., 2022). These include changes in community composition from adaptation and evolution (e.g., Boyd, 2015, Passow & Carlson, 2012, Doney et al., 2009, Lomas et al., 2022), changes in nitrogen and/or iron limitation (e.g., Thornton et al., 2009, Jickells et al., 2005), or changes in early diagenesis and sediment fluxes (e.g., Griffiths et al., 2017; Sweetman et al., 2017). However, at least some of our model's mechanistic shortcomings are partially compensated by having optimized biogeochemical parameters (Pasquier et al., 2023).

*L. 181: I suggest rephrasing "The future circulation of our states".*
We have replaced with "The circulation of our future states".

*L. 185-187: I suggest showing the nutrient supply and its changes instead of (or in addition to) the euphotic nutrient inventory and its changes, which result from several processes.*
The euphotic-zone nutrient inventory may be considered to be the nutrient *supply* as nutrient uptake rates depend on nutrient concentration and not nutrient flux. While one could consider the net nutrient flux into the euphotic zone, this flux also results from a number of processes. We think quantifying the standing stock of nutrients if preferable to make our points here. No change.

*L 192: Maybe better "and despite" instead of "despite".*
Done.

*L. 192-200: Please see my first major comment. The authors presented that the main changes in mean euphotic nutrient concentration are localized in areas where ML is unrealistically deep in the preindustrial simulation (Fig. B1d-e, Fig. C2). Could the authors, in the discussion section, discuss or indicate an estimate of the uncertainties of the different contributions to DIC biological uptake changes and of the resulting uptake change, associated with the MLD biases, perhaps based on previous studies? Could the MLD biases impact the sign of the total production change and the first main conclusion?*
As per our general response at the beginning, quantitative estimates of such uncertainty are out of scope here. However, our qualitative results are likely to be robust as the Southern Ocean mixed layer is predicted to shoal (de Lavergne et al., 2014; Kwiatkowski et al., 2020), driving a decline in nutrient supply, which in steady state must be balanced by a decline in export (and likely production).
Yes, a smaller change in MLD could impact the sign of the production change, whereby the increase in phytoplankton growth from warming could overtake the decline in nutrients. However, our key point here is that the regenerated carbon inventories increase because of the increased residence times, even if production declines. Also note that comparisons with transient models are complicated, with most CMIP5 and CMIP6 models predicting an increase in Southern Ocean production by 2100, potentially caused by any combination of a warming-enhanced growth, reduced zooplankton grazing pressure, increases in micronutrient supply, reduced sea ice, or reduced light stress from increased stratification and a shallower MLD (Kwiatkowski et al., 2020). In response, we have added the following to the caveats in Section 4 (Discussion):

> Although a model with more realistic AABW formation might produce quantitatively different results (e.g., production could increase due to a smaller decline in mixed layer depths), the qualitative links between changes and their driving mechanisms should be robust and model-independent.

*L. 201: Maybe better: "Changes in export ratio"*
Thank you. We have adopted this wording.

*L. 202-203: I suggest defining export production at the beginning of "Changes in export ratios" section, instead of in L. 217-218.*
Agreed. We have moved the definition of export production.

*L. 208-215: In this paragraph, the authors listed mechanisms inducing changes in OC export ratio. Did the authors quantify the changes of DOC and POC export ratios, DOC and POC exports, and euphotic DOC and POC remineralizations? I suggest adding these changes to the appendices. Related to my first main comment, could the authors give an estimate of the uncertainties of the magnitude of export ratio changes due the MLD biases?*

Yes, we quantified changes in export ratios for each export pathway (DOC, $POC_s$, $POC_f$, and PIC) but, for concision, we only showed maps of the total export ratio in the main text. However, Appendix Fig. D3 does show the corresponding zonal integrals partitioned according to export pathway, which we think provides ample detail. As per our overall response at the beginning, quantifying the uncertainty is not feasible within the scope of our study. In response, we now refer to Figure D3 in this passage.

*L. 217: "changes in carbon export production Jex itself".*
OK, we have added "production" here.

*L. 223: Do you mean organic-matter production or export production? Please clarify.*
We meant export production. We have added "export".

*L. 230: "suggest"*
Fixed (removed erroneous "s" at the end).

*L. 259: "unrealistic deep ML"*
We have changed this for "unrealistically deep mixed layer".

*L. 280-281: I suggest rephrasing this sentence.*
While we are not exactly sure what Referee #2 is suggesting here to read:

> Figure 5 shows that for each export mechanism (DOC, $POC_s$, $POC_f$, and PIC), export production declines and sequestration time increases. The increases in sequestration time are consistent with an overall slowdown of the circulation and with longer re-exposure times for regenerated DIC to return to the euphotic zone (Fig. C3).

*L. 302: "these pathways change"*
We think it is clear that "these" refers to the DIC pathways in this short sentence. No change.

*L. 303: delete [t]*
Apologies. Deleted.

*L. 338: Do you mean "regenerated nutrients at intermediate depths"?*
We have rephrased as "nutrients regenerated at intermediate depths".

*L. 345-346: I suggest deleting the repetitions with L. 253-254, if any.*
We are not sure where the repetition is here. L345–346 is about the preformed carbon inventories while L253–254 is about the total carbon inventories. No change.

*L. 367-368: the flow rate decreases/increases or the flow rate slows/speeds up?*
We have removed "rate" here and made sure to use "the flow slows/speeds up".

*L. 391-393: I suggest replacing "a significant advance over" by "an enrichment of"*
We respectfully disagree here. The traditional view of preformed tracers as the solution to a concentration boundary-value problem is set in stone and cannot be "enriched" by going to flow rates (i.e., source/sink). No change.

*L. 449-453: "the changes in the controls on carbon export and biological utilization identified by Boyd (2015)": Could the authors list these changes and clarify the agreements with the study of Boyd et al. (2015)?*
There are 12 carbon pump components identified by Boyd (2015) that are based on 25 publications (their Table 3). In our judgement, an exhaustive comparison would be too much detail for this discussion. For simplicity and clarity, we instead focused on the strongest disagreement, which is that PCO2 does not include shifts in community composition, which is the predominant control in the study by Boyd (2015). No change.

*Appendices: I suggest (i) numbering the appendices and their figures in the order of reference in the main text, (ii) locating the figures after the title of the corresponding appendix (Fig. C1),*

*(iii) changing the caption of Figure D2 (and Fig. D1), where the authors referred to Fig. 2 that is commented later in the main text.*

We respectfully disagree with Referee #2 here. The appendices were organized for clarity and ordered by the first appearance of the first figure of each appendix in the main text: Appendix A is mentioned in L170, Fig. B1 in L187, Fig. C1–3 in L190, and Fig. D1–3 in L203. In response, figure placement with respect to titles has been fixed, although the precise placement of figures will occur in the final typesetting by the journal. We have also revised the captions of appendix figures so that they are self-contained.

*L. 533-535 and L. 613-618 : These lines should be located after the appendices.*

We used the Journal's LaTeX template, which controls these placements. We defer to the editors for final typesetting details. No change.

*Figure C2: I suggest adding the MLD changes for the two future scenarios as in Figure 2.*
Agreed. We have added plots of the changes to Fig. C2.

**References (including those already cited in the manuscript)**

Arias, P., Bellouin, N., Coppola, E., Jones, R., Krinner, G., Marotzke, J., Naik, V., Palmer, M., Plattner, G.-K., Rogelj, J., Rojas, M., Sillmann, J., Storelvmo, T., Thorne, P., Trewin, B., Achuta Rao, K., Adhikary, B., Allan, R., Armour, K., ... Zickfeld, K. (2021). Technical Summary. In V. Masson-Delmotte, P. Zhai, A. Pirani, S. Connors, C. Péan, S. Berger, N. Caud, Y. Chen, L. Goldfarb, M. Gomis, M. Huang, K. Leitzell, E. Lonnoy, J. Matthews, T. Maycock, T. Waterfield, O. Yelekçi, R. Yu, & B. Zhou (Eds.), *Climate Change 2021: The Physical Science Basis. Contribution of Working Group I to the Sixth Assessment Report of the Intergovernmental Panel on Climate Change* (pp. 33–144). Cambridge University Press. https://doi.org/10.1017/9781009157896.002

Bao, Y., & Li, Y. (2016). Simulations of Dissolved Oxygen Concentration in CMIP5 Earth System Models. *Acta Oceanologica Sinica*, *35*(12), 28–37. https://doi.org/10.1007/s13131-016-0959-x

Bardin, A., Primeau, F. W., & Lindsay, K. (2014). An offline implicit solver for simulating prebomb radiocarbon. *Ocean Modelling*, *73*, 45–58. https://doi.org/10.1016/j.ocemod.2013.09.008

Bi, D., Marsland, S. J., Uotila, P., O'Farrell, S., Fiedler, R. A. S., Sullivan, A., Griffies, S. M., Zhou, X., & Hirst, A. C. (2013). ACCESS-OM: The Ocean and Sea Ice Core of the ACCESS Coupled Model. *Australian Meteorological and Oceanographic Journal*, *63*, 213–232.

Boyd, P. W. (2015). Toward quantifying the response of the oceans' biological pump to climate change. *Frontiers in Marine Science*, *2*. https://doi.org/10.3389/fmars.2015.00077

Chamberlain, M. A., Matear, R. J., Holzer, M., Bi, D., & Marsland, S. J. (2019). Transport matrices from standard ocean-model output and quantifying circulation response to climate change. *Ocean Modelling*, *135*, 1–13. https://doi.org/10.1016/j.ocemod.2019.01.005

de Lavergne, C., Palter, J. B., Galbraith, E. D., Bernardello, R., & Marinov, I. (2014). Cessation of Deep Convection in the Open Southern Ocean under Anthropogenic Climate Change. *Nature Climate Change*, *4*(4), 278–282. https://doi.org/10.1038/nclimate2132

DeVries, T., & Holzer, M. (2019). Radiocarbon and Helium Isotope Constraints on Deep Ocean Ventilation and Mantle-\textsuperscript{3}He Sources. *Journal of Geophysical Research: Oceans*, *124*(5), 3036–3057. https://doi.org/10.1029/2018JC014716

Doney, S. C., Fabry, V. J., Feely, R. A., & Kleypas, J. A. (2009). Ocean Acidification: The Other CO\textsubscript{2} Problem. *Annual Review of Marine Science*, *1*(1), 169–192. https://doi.org/10.1146/annurev.marine.010908.163834

Drenkard, E. J., John, J. G., Stock, C. A., Lim, H.-G., Dunne, J. P., Ginoux, P., & Luo, J. Y. (2023). The Importance of Dynamic Iron Deposition in Projecting Climate Change Impacts on Pacific Ocean Biogeochemistry. *Geophysical Research Letters*, *50*(21), e2022GL102058. https://doi.org/10.1029/2022GL102058

Fu, W., Moore, J. K., Primeau, F., Collier, N., Ogunro, O. O., Hoffman, F. M., & Randerson, J. T. (2022). Evaluation of Ocean Biogeochemistry and Carbon Cycling in CMIP Earth System Models With the International Ocean Model Benchmarking (IOMB) Software System. *Journal of Geophysical Research: Oceans*, *127*(10), e2022JC018965. https://doi.org/10.1029/2022JC018965

Griffiths, J. R., Kadin, M., Nascimento, F. J. A., Tamelander, T., Törnroos, A., Bonaglia, S., Bonsdorff, E., Brüchert, V., Gårdmark, A., Järnström, M., Kotta, J., Lindegren, M., Nordström, M. C., Norkko, A., Olsson, J., Weigel, B., Żydelis, R., Blenckner, T., Niiranen, S., & Winder, M. (2017). The Importance of Benthic--Pelagic Coupling for Marine Ecosystem Functioning in a Changing World. *Global Change Biology*, *23*(6), 2179–2196. https://doi.org/10.1111/gcb.13642

Henson, S. A., Laufkötter, C., Leung, S., Giering, S. L. C., Palevsky, H. I., & Cavan, E. L. (2022). Uncertain Response of Ocean Biological Carbon Export in a Changing World. *Nature Geoscience*, *15*(4), 248–254. https://doi.org/10.1038/s41561-022-00927-0

Holzer, M., Chamberlain, M. A., & Matear, R. J. (2020). Climate-Driven Changes in the Ocean's Ventilation Pathways and Time Scales Diagnosed From Transport Matrices. *Journal of Geophysical Research: Oceans*, *125*(10), e2020JC016414. https://doi.org/10.1029/2020JC016414

Holzer, M., Kwon, E. Y., & Pasquier, B. (2021). A New Metric of the Biological Carbon Pump: Number of Pump Passages and Its Control on Atmospheric pCO\textsubscript{2}. *Global Biogeochemical Cycles*, *35*(6), e2020GB006863. https://doi.org/10.1029/2020GB006863

Huang, Q., Primeau, F. W., & DeVries, T. (2021). CYCLOCIM: A 4-D variational assimilation system for the climatological mean seasonal cycle of the ocean circulation. *Ocean Modelling*, *159*, 101762. https://api.semanticscholar.org/CorpusID:234316061

Ito, T., Bracco, A., Deutsch, C., Frenzel, H., Long, M., & Takano, Y. (2015). Sustained growth of the Southern Ocean carbon storage in a warming climate. *Geophysical Research Letters*, *42*(11), 4516–4522. https://doi.org/10.1002/2015GL064320

Jickells, T. D., An, Z. S., Andersen, K. K., Baker, A. R., Bergametti, G., Brooks, N., Cao, J. J., Boyd, P. W., Duce, R. A., Hunter, K. A., Kawahata, H., Kubilay, N., laRoche, J., Liss, P. S., Mahowald, N., Prospero, J. M., Ridgwell, A. J., Tegen, I., & Torres, R. (2005). Global Iron Connections Between Desert Dust, Ocean Biogeochemistry, and Climate. *Science*, *308*(5718), 67–71. https://doi.org/10.1126/science.1105959

Kwiatkowski, L., Torres, O., Bopp, L., Aumont, O., Chamberlain, M., Christian, J. R., Dunne, J. P., Gehlen, M., Ilyina, T., John, J. G., Lenton, A., Li, H., Lovenduski, N. S., Orr, J. C., Palmieri, J., Santana-Falcón, Y., Schwinger, J., Séférian, R., Stock, C. A., ... Ziehn, T. (2020). Twenty-First Century Ocean Warming, Acidification, Deoxygenation, and Upper-Ocean Nutrient and Primary Production Decline from CMIP6 Model Projections. *Biogeosciences*, *17*(13), 3439–3470. https://doi.org/10.5194/bg-17-3439-2020

Li, Q., England, M. H., Hogg, A. M., Rintoul, S. R., & Morrison, A. K. (2023). Abyssal Ocean Overturning Slowdown and Warming Driven by Antarctic Meltwater. *Nature*, *615*(7954), 841–847. https://doi.org/10.1038/s41586-023-05762-w

Liu, J., Wang, X., Wu, D., Wei, H., Li, Y., & Ji, M. (2024). Historical Footprints and Future Projections of Global Dust Burden from Bias-Corrected CMIP6 Models. *Npj Climate and Atmospheric Science*, *7*(1), 1–12. https://doi.org/10.1038/s41612-023-00550-9

Liu, Y., Moore, J. K., Primeau, F. W., & Wang, W. L. (2023). Reduced CO\textsubscript{2} Uptake and Growing Nutrient Sequestration from Slowing Overturning Circulation. *Nature Climate Change*, *13*(1), 83–90. https://doi.org/10.1038/s41558-022-01555-7

Lomas, M. W., Bates, N. R., Johnson, R. J., Steinberg, D. K., & Tanioka, T. (2022). Adaptive Carbon Export Response to Warming in the Sargasso Sea. *Nature Communications*, *13*(1), 1211. https://doi.org/10.1038/s41467-022-28842-3

Meinshausen, M., Smith, S. J., Calvin, K., Daniel, J. S., Kainuma, M. L. T., Lamarque, J.-F., Matsumoto, K., Montzka, S. A., Raper, S. C. B., Riahi, K., Thomson, A., Velders, G. J. M., & van Vuuren, D. P. (2011). The RCP Greenhouse Gas Concentrations and Their Extensions from 1765 to 2300. *Climatic Change*, *109*(1), 213. https://doi.org/10.1007/s10584-011-0156-z

Pasquier, B., Holzer, M., Chamberlain, M. A., Matear, R. J., Bindoff, N. L., & Primeau, F. W. (2023). Optimal parameters for the ocean's nutrient, carbon, and oxygen cycles compensate for circulation biases but replumb the biological pump. *Biogeosciences*, *20*(14), 2985–3009. https://doi.org/10.5194/bg-20-2985-2023

Passow, U., & Carlson, C. A. (2012). The Biological Pump in a High CO2 World. *Marine Ecology Progress Series*, *470*, 249–271. https://doi.org/10.3354/meps09985

Planchat, A., Kwiatkowski, L., Bopp, L., Torres, O., Christian, J. R., Butenschön, M., Lovato, T., Séférian, R., Chamberlain, M. A., Aumont, O., Watanabe, M., Yamamoto, A., Yool, A., Ilyina, T., Tsujino, H., Krumhardt, K. M., Schwinger, J., Tjiputra, J., Dunne, J. P., & Stock, C. (2023). The Representation of Alkalinity and the Carbonate Pump from CMIP5 to CMIP6 Earth System Models and Implications for the Carbon Cycle. *Biogeosciences*, *20*(7), 1195–1257. https://doi.org/10.5194/bg-20-1195-2023

Purich, A., & England, M. H. (2023). Projected Impacts of Antarctic Meltwater Anomalies over the Twenty-First Century. *Journal of Climate*, *36*(8), 2703–2719. https://doi.org/10.1175/JCLI-D-22-0457.1

Purich, A., England, M. H., Cai, W., Sullivan, A., & Durack, P. J. (2018). Impacts of Broad-Scale Surface Freshening of the Southern Ocean in a Coupled Climate Model. *Journal of Climate*, *31*(7), 2613–2632. https://doi.org/10.1175/JCLI-D-17-0092.1

Riahi, K., van Vuuren, D. P., Kriegler, E., Edmonds, J., O'Neill, B. C., Fujimori, S., Bauer, N., Calvin, K., Dellink, R., Fricko, O., Lutz, W., Popp, A., Cuaresma, J. C., KC, S., Leimbach, M., Jiang, L., Kram, T., Rao, S., Emmerling, J., … Tavoni, M. (2017). The Shared Socioeconomic Pathways and their energy, land use, and greenhouse gas emissions implications: An overview. *Global Environmental Change*, *42*, 153–168. https://doi.org/10.1016/j.gloenvcha.2016.05.009

Riebesell, U., Körtzinger, A., & Oschlies, A. (2009). Sensitivities of marine carbon fluxes to ocean change. *Proceedings of the National Academy of Sciences*, *106*(49), 20602–20609. https://doi.org/10.1073/pnas.0813291106

Sweetman, A. K., Thurber, A. R., Smith, C. R., Levin, L. A., Mora, C., Wei, C.-L., Gooday, A. J., Jones, D. O. B., Rex, M., Yasuhara, M., Ingels, J., Ruhl, H. A., Frieder, C. A., Danovaro, R., Würzberg, L., Baco, A., Grupe, B. M., Pasulka, A., Meyer, K. S., … Roberts, J. M. (2017). Major Impacts of Climate Change on Deep-Sea Benthic Ecosystems. *Elementa: Science of the Anthropocene*, *5*, 4. https://doi.org/10.1525/elementa.203

Tagliabue, A., Aumont, O., & Bopp, L. (2014). The Impact of Different External Sources of Iron on the Global Carbon Cycle. *Geophysical Research Letters*, *41*(3), 920–926. https://doi.org/10.1002/2013GL059059

Taylor, K., Stouffer, R., & Meehl, G. (2012). An Overview of CMIP5 and the experiment design. *Bull. Amer. Meteor. Soc.*, *93*, 485–498. https://doi.org/10.1175/BAMS-D-11-00094.1

Thornton, P. E., Doney, S. C., Lindsay, K., Moore, J. K., Mahowald, N., Randerson, J. T., Fung, I., Lamarque, J.-F., Feddema, J. J., & Lee, Y.-H. (2009). Carbon-nitrogen interactions regulate climate-carbon cycle feedbacks: results from an atmosphere-ocean general circulation model. *Biogeosciences*, *6*(10), 2099–2120. https://doi.org/10.5194/bg-6-2099-2009

Volk, T., & Hoffert, M. I. (1985). Ocean Carbon Pumps: Analysis of Relative Strengths and Efficiencies in Ocean-Driven Atmospheric CO\textsubscript{2} Changes. In E. T. Sundquist & W. S. Broecker (Eds.), *The Carbon Cycle and Atmospheric CO\textsubscript{2}: Natural Variations Archean to Present* (pp. 99–110). American Geophysical Union (AGU). https://doi.org/10.1029/GM032p0099

---

## Author Response (AR2)

**Responses to Reviewers' comments**

**manuscript**: egusphere-2023-2525
**journal**: Biogeosciences
**title**: The ocean's biological and preformed carbon pumps in perpetually slower and warmer oceans
**authors**: Benoît Pasquier, Mark Holzer, Matthew A. Chamberlain

**Response to Reviewer 1**

**Note:** Reviewer 1 comments are indicated in black and our responses are in blue.

The authors improved the manuscript and answered to the comments and questions of the reviewers. I recommend publication in Biogeosciences after addressing the minor comments below. The line numbers correspond to the ones in the revised manuscript with track changes.

We thank Reviewer 1 for the positive feedback and for recommending our manuscript for publication in Biogeosciences. The minor comments are addressed below.

I understand that ACCESS1.3 is state-of-the art climate model and that previous publication optimized the PCO2 biogeochemistry model embedded in the ACCESS1.3 circulation. The choice of this model is clearly justified for this study. Regarding the influence of meltwater input from terrestrial ice sheets on the ventilation and storage of carbon in the deep ocean, the authors added in the discussion of the revised manuscript that they expect an exacerbated reduction in future Southern Ocean ventilation from the ice-sheet meltwater. The future influence of meltwater, and especially Antarctic meltwater, on regional and global climate under RCP4.5 or RCP8.5 scenarios is likely to result in strong Southern Ocean changes. The authors mentioned the ACCESS-ESM1.5 model that was used in Purich and England (2023) to investigate the effect of ice-sheet meltwater in the Southern Ocean. ACCESS-ESM1.5 is based on ACCESS1.3, so I wonder if it would have been possible to investigate the effect of ice-sheet meltwater using this model instead of ACCESS1.3? Or otherwise via sensitivity experiments?

We agree that the influence of ice-sheet meltwater will be very important in the future, especially in the Southern Ocean. Given that ACCESS-ESM1.5 is essentially an updated version of ACCESS1.3, it is appropriate to discuss the experiments by Purich and England (2023) in our Discussion section, and we think this is sufficient to highlight the issue. Repeating our analyses for a circulation from ACCESS-ESM1.5 (with and without ice-sheet melt-water effects) is well out of scope here as it would require constructing the appropriate transport

matrix (including tuning mixing and validating transport), not to mention additional analysis and interpretation in the context of our frozen-in-time steady-state framework. (No changes to the manuscript in response to this comment.)

Section 3.1.2 discusses how warming stimulates euphotic POC respiration and affects particle sinking rates. This section could provide a clearer distinction between the direct and indirect effects of warming on these processes.

We agree that our discussion of these effects could have been clearer. In response, we have revised the last paragraph of Section 3.1.2 to start with:

> In PCO2, warming has opposing effects on the key mechanisms controlling export efficiency. On one hand, warming stimulates respiration of POC in the euphotic zone leaving less POC to be exported, which tends to decrease export ratios. On the other hand, warming reduces water viscosity so that particles sink faster out of the euphotic zone, which tends to increase export ratios (see Eqs. (2) and (A3) in Pasquier et al., 2023).

L210: please detail the change in C:P uptake ratio

C:P uptake ratio increases when phosphate concentration decreases in the parameterization of Eq. (2) that we adopted from Galbraith and Martiny (2015). In response, we have added a reference to Eq. (2), and the revised sentence now reads:

> The decreased nutrient supply (i.e., reduced phosphate concentration) reduces biological carbon uptake despite increasing the C:P uptake ratio as parameterized by Eq. (2) following Galbraith and Martiny (2015).

L306: relative to complete surface saturation

Thanks for catching this typo. We have corrected "realtive".

L386: Note that if we had

We agree that the suggested wording is better. We have replaced "Note that, had we" with "Note that if we had".

**Response to Reviewer 2**

**Note:** Reviewer 2 comments are indicated in black and our responses are in blue.

The authors have addressed my previous comments in their answer and the new version of the manuscript is now much improved. In their revision, the authors have clarified the

methodology and expanded the discussion of their results. There is just a minor point regarding the timescales considered in this study which I think should be addressed. The authors state that their steady state study captures the response of the system on all timescales. I think it could lead to confusion (as discussed by the authors, some timescales are not taken into account in the biogeochemical simulations, for example seasonal timescales) and I suggest clarifying this point in the manuscript. Besides, I have noticed a few additional minor and technical comments on specific sections of the text.

Overall, the paper has novel aspects with a partitioning of preformed and regenerated dissolved inorganic carbon in accounting for the source and sink processes and it makes a valuable contribution to the understanding of carbon pump and its changes under two climate change scenarios. I look forward to seeing the final manuscript published after these final issues are addressed.

We thank Reviewer 2 for the positive feedback and for the additional comments.

A distinction must be made between the timescales of variability (e.g., seasonal) and the timescales of the steady-state system (e.g., decadal thermocline ventilation and centennial deep ventilation). We agree that our framework does not capture variability on any timescale, as we focus on steady-state biogeochemistry in frozen-in-time circulations. However, our model does capture the response of the system to changes in decadal-mean ocean state (preindustrial to perpetual 2090s) integrated over all the timescales of the system. We have clarified this distinction throughout (including in the Abstract). Our point-by-point responses to the minor comments follow below.

Please see all my detailed and technical comments below.

L. 4-6: "Focusing on steady-state changes from preindustrial conditions allows us to capture the response of the system on all timescales, not just on the sub-centennial timescales of typical transient simulations." I don't think it allows capturing the response of the system on all timescales and I suggest this sentence needs revising.

To clarify that our analysis captures the system's response integrated over all timescales, we have revised this passage to read:

> Focusing on steady-state changes from preindustrial conditions allows us to capture the response of the system integrated over all the timescales of the steady-state biogeochemistry, as opposed to typical transient simulations that capture only sub-centennial timescales.

L. 51: I suggest removing they probe all the timescales of the system and.

While our frozen-in-time steady-state framework cannot capture natural variability on any timescale, we do capture the system's response to change in ocean state integrated over all timescales as discussed above. In response, we have revised the passage as follows:

> Probing the system in steady state is advantageous because it avoids the complications of transience by integrating the system's response to change in perpetual decadal-mean ocean state on all timescales.

L. 74: change analysed to analyzed, as used throughout the manuscript.

Thanks for catching this. Done.

L. 94: I suggest rephrasing this sentence as for example: Instead, our steady-state solutions allow us to determine what the asymptotically long-term

We have rephrased as suggested.

L. 141: I suggest defining r

Thank you for the suggestion. The revised sentence now reads:

> Note that in our model, carbon can only enter or exit the ocean through air–sea exchange so that in steady-state equilibrium there is no net carbon source or sink when globally integrating over all locations $r$, i.e., $\int J_{\mathrm{atm}}(r)\, \mathrm{d}^3 r = 0$.

L. 219-220: In response to a previous comment, the authors have added here a sentence in which they refer to Fig. D3. The contributions from each export pathway shown on this figure are described in the next sub-sections. I suggest moving this sentence in one of the next sub-sections and here referring to this next sub-section, or, at least, guiding the reading of this figure, by indicating, for example, the relevant contributions (and the colour of the corresponding curves).

We respectfully disagree that this suggestion would improve readability, with the order of presentation of the various aspects of export is somewhat arbitrary. While the contributions from each export pathway to global export and regenerated C inventories are described later (Fig. 5), the colored curves in Fig. D3 show the latitudinal decomposition of $\Delta J_{\mathrm{ex}}$ for each export pathway into sub-contributions from changes in uptake and changes in export ratios, i.e., from each of $f\Delta U$, $U\Delta f$, and $\Delta f\Delta U$. In addition, the globally integrated $\Delta J_{\mathrm{ex}}$ from each pathway and for each scenario were already shown in the panels of Figure D3 as indicated in its caption. Adding the $4 \times 8 = 32$ values of the globally-integrated sub-contributions to the text or the figure would make them very cluttered for little additional insight. (No change to the manuscript in response to this comment.)

L. 559-562: I suggest rephrasing the last part of the sentence.

Agreed. The sentence now reads:

> While the responses of the biological and preformed pumps are driven by different mechanisms with widely different response timescales, the regenerated and preformed DIC inventories both increase by similar absolute amounts when the contributions from the longest response timescales are captured in steady state.

**Additional note from the authors**

As emphasized in the previous round of revisions, our idealized framework of steady-state biogeochemistry embedded in frozen-in-time ocean states precludes the interpretation of our results as predictions of the future. During the review of a separate manuscript on ocean deoxygenation using the same framework, a reviewer made us aware that our use of the term "future" could be confusing in this regard. We have therefore clarified in this revision that our results quantify the carbon pumps in perpetually warmer and slower oceans. Specifically, we have replaced "future" with "warmer and slower" or "perpetual 2090s" throughout the manuscript, including title, abstract, and figures. We have also revised the first caveat in the Discussion section to read as follows:

> Our results for steady-state biogeochemistry embedded in frozen-in-time ocean states are not predictions of the future. The real ocean will keep changing for many centuries beyond the 2090s and its future long-term dynamical equilibrium (if forcing ever stabilizes) will likely be vastly different from the ocean states analyzed here (e.g., Schmittner et al., 2008). However, our idealized steady states do reveal the key mechanisms driving the system's responses to future change.

We think these revisions will prevent potential confusion and help readers to better understand the nature of our analysis and the implications of our results.

**References**

Galbraith, E. D., & Martiny, A. C. (2015). A simple nutrient-dependence mechanism for predicting the stoichiometry of marine ecosystems. *Proceedings of the National Academy of Sciences*, *112*(27), 8199–8204. https://doi.org/10.1073/pnas.1423917112

Pasquier, B., Holzer, M., Chamberlain, M. A., Matear, R. J., Bindoff, N. L., & Primeau, F. W. (2023). Optimal parameters for the ocean's nutrient, carbon, and oxygen cycles compensate for circulation biases but replumb the biological pump. *Biogeosciences*, *20*(14), 2985–3009. https://doi.org/10.5194/bg-20-2985-2023

Purich, A., & England, M. H. (2023). Projected Impacts of Antarctic Meltwater Anomalies over the Twenty-First Century. *Journal of Climate*, *36*(8), 2703–2719. https://doi.org/10.1175/JCLI-D-22-0457.1

Schmittner, A., Oschlies, A., Matthews, H. D., & Galbraith, E. D. (2008). Future changes in climate, ocean circulation, ecosystems, and biogeochemical cycling simulated for a business-as-usual $CO_2$ emission scenario until year 4000 ad. *Global Biogeochemical Cycles*, *22*(1). https://doi.org/10.1029/2007GB002953